# An integrative approach to the study of *Kempnyia* Klapálek, 1914 (Plecoptera: Perlidae) from Brazil: Support for the description of four new species and a basis for future studies

**Lucas Henrique de Almeida**[1]*, **Maísa de Carvalho Gonçalves**[2], **Pitágoras da Conceição Bispo**[1]

**1** Laboratório de Biologia Aquática, Faculdade de Ciências e Letras de Assis, Universidade Estadual Paulista, Assis, São Paulo, Brazil, **2** PPG em Entomologia, Universidade Federal de Viçosa (UFV), Viçosa, Minas Gerais, Brazil

* lucasalmeida768@yahoo.com.br

**Data Availability Statement:** All COI sequences generated in the present study are available in

## Abstract

*Kempnyia* (Plecoptera: Perlidae) is an endemic genus of Brazilian stoneflies that has 36 valid species and is distributed primarily in the Atlantic Forest and the mountainous areas of Central Brazil, particularly in Goiás and Tocantins states. Despite being the Brazilian genus with the most DNA sequences available on GenBank, integrative studies on the genus began only recently, in 2014. In this context, herein we studied the morphology and molecular data of *Kempnyia* specimens deposited in the Aquatic Biology Laboratory (UNESP, Assis) and the Entomology Museum of the Federal University of Viçosa (UFVB, Viçosa) collections. For the integrative approach adopted, in addition to studying the specimens morphologically, we used sequences of the COI mitochondrial gene combined with the following species delimitation methods: Automatic Barcode Gap Discovery (ABGD), both primary (ABGD$_p$) and recursive (ABGD$_r$) partitions; Assemble Species by Automatic Partitioning (ASAP); Poisson Tree Processes (PTP) and the Bayesian implementation of the Poisson Tree Processes (bPTP). As a result, we provided 28 new COI sequences of 21 species and support the description of four new species, namely, *K. guarani* **sp. nov**., *K. tupiniquim* **sp. nov**., *K. una* **sp. nov**., and *K. zwickii* **sp. nov**., consequently increasing the known diversity of the genus to 40 species. We also discuss the morphological variations observed in other species of the genus and provide several new geographic records. Therefore, our study brings new insights into the values of intra- and interspecific molecular divergence within *Kempnyia*, serving as a basis for new studies.

## Introduction

Plecoptera (stoneflies) are an order of aquatic insects that play an important role as indicators of water quality due to their sensitivity to environmental changes and pollution, being widely

GenBank with the accession codes available in the supplementary material.

**Funding:** All authors: Research Foundation of São Paulo (FAPESP) (grants 2019/22833-0 and BIOTA 2021/05986-8) and National Council for Scientific and Technological Development (CNPq) (grant PROTAX 441119/2020-4). LHA and MCG: Coordination for the Improvement of Higher Education Personnel (CAPES). LHA: São Paulo State University (UNESP) and PROTAX-CNPq (grant 150032/2024-2). PCB: CNPq (grant 306400/2022-7).

**Competing interests:** The authors have declared that no competing interests exist.

used in the biomonitoring of rivers and streams and ecological studies [1]. The order has more than 3700 species and 17 families found throughout the world, with the exception of Antarctica [2, 3]. Perlidae Latreille 1802 [4] emerged about 165 Mya during the Jurassic period in Laurasia [5] and today is the most diverse family among Plecoptera, with at least 1100 described species recorded in all suitable regions of the world, except the Australasia region [2]. In the Neotropics [6], Pessacq *et al*. [7] counted about 450 valid species, representing more than 70% of all diversity of stoneflies in this region. Among them, about 150 are recorded in Brazil and belong to the genera *Anacroneuria* Klapálek 1909 [8], *Enderleina* Jewett 1960 [9], *Kempnyia* Klapálek 1914 [10], and *Macrogynoplax* Enderlein 1909 [11].

Klapálek [10] proposed the genus *Kempnyia*, describing *K. tenebrosa* Klapálek 1916 [12] as the type species [13]. Currently, it encompasses the species formerly placed in the old genera *Eutactophebia* Klapálek 1914 [10], *Collampla* Navás 1929 [14], *Diperla* Navás 1936 [15], *Forca* Navás 1925 [16], *Laessia* Navás 1934 [17], and *Nedanta* Navás 1932 [9, 18–22]. The genus is endemic to Brazil, has 36 valid species [7, 23] and is distributed mainly along the Atlantic Forest and in the mountainous regions of Central Brazil, especially in Goiás and Tocantins states [7, 24]. *Kempnyia* are medium to large stoneflies, with the size of the male's forewing varying from 8 to more than 24 mm [13]. Living adults have the color ranging from bright yellow to ochraceous and black, and in some cases may present different patterns of spots on the head and pronotum [23].

Zwick [20, 25, 26], Froehlich [21, 27, 28], and Dorvillé & Froehlich [29] studied several older species of *Kempnyia*, and more recently new species were described by Bispo & Froehlich [30], Froehlich [31, 32], and Avelino-Capistrano *et al*. [33–35]. Despite this, the genus presents several taxonomic problems, mainly involving species with old and insufficient descriptions. Among the 36 species described: 23 were based on both males and females; eight only on males, and five only on females. Only nine of these 36 species have nymphs associated and described and least nine species have problematic descriptions and holotypes deposited in museums outside Brazil, with some of them declared missing or lost [36].

One of the most discussed alternatives to help solve these problems is the use of an integrative taxonomic approach. The first works addressing the integration of molecular and morphological information to study Perlidae in Brazil were dedicated to the genus *Kempnyia* [34, 35]. The first intended to associate and describe immatures of the genus, resulting in the description of two nymphs not yet known by name [34], while the second used sequences from a few different species to support the description of a new one, also leading to the association and description of the respective immature [35]. However, it was only in 2020 that new sequences were published in a study dedicated to the Plecoptera fauna from the Paranapiacaba Mountains, in São Paulo state. As a result, *Kempnyia* is the Brazilian perlid genus with the greatest diversity of sequences available on GenBank, having 37 sequence deposits from 10 species.

In the present study, adult specimens of *Kempnyia* from different localities were studied under an integrative approach with the aim of reducing the Linnean and Wallacean shortfalls [37] with the description of new species and the expansion of knowledge about the geographic occurrence of species, respectively. Additionally, comments on the morphological and molecular variations of the species were included and new barcode sequences (COI) of *Kempnyia* species were deposited in GenBank.

## Methods

All materials were collected according to Brazilian legislation: using collection licenses from the Authorization and Information System in Biodiversity (SISBIO– 55428–16 and 65213–11);

the Environmental Research Institute of São Paulo State (IPA/SIMA–SMA 15054/2022 and 5579/2023), and the Secretariat for the Environment and Sustainable Development of Goiás State (202100017008515). Furthermore, animal handling was carried out in accordance with the normative instructions of the Chico Mendes Institute for Biodiversity Conservation (ICMBio).

The specimens were attracted and sampled using a traditional white sheet and a light pan trap, both with ultraviolet and white lamps [38] or with blue, green and ultraviolet LEDs [39]. The materials collected before 2014 were preserved in 80% ethanol at air-conditioned temperature (17 °C). In the case of the materials collected from 2014, the insects were stored in absolute ethanol in a freezer (-25 °C) so as to better preserve their DNA. The specimens studied were deposited in the Aquatic Biology Laboratory Colletion (CLBA—UNESP, Assis), the Entomology Museum of the Federal University of Viçosa (UFVB, Viçosa), and the Museum of Zoology of the University of São Paulo (MZUSP, São Paulo). The holotypes and paratypes of the four new species were deposited in the CLBA and UFVB. The main collectors, organized according to their last names, were: Almeida–Lucas Henrique de Almeida; Avelino-Capistrano: Fernanda Avelino-Capistrano; Bispo–Pitágoras da Conceição Bispo; Blahnik–Roger Blahnik; Cabette–Helena Soares Ramos Cabette; Calor–Adolfo Calor; Campos–Rogério Campos; Dias–Everton Santos Dias; Froehlich–Claudio Gilberto Froehlich; Fusari–Lívia Maria Fusari; Melo–Adriano Sanches Melo; Lecci–Lucas Lecci; Paprocki–Henrique Paprocki; Miguel–Marina Miguel; Pinho–Luiz Carlos de Pinho; Roque–Fabio de Oliveira Roque; and Salles–Frederico F. Salles.

For identification, the abdomen of the males were severed between segments seven and eight and treated overnight with 10% potassium hydroxide (KOH) to clear the tissues. In order to neutralize the reaction, the abdomens were then placed in acetic acid and washed with 80% ethanol. The extraction of penial armature was made to identify the species based on the morphological comparison of the penial armature already described in the literature, mainly the studies developed by Zwick [20, 25, 26], Froehlich [21, 27, 28, 31, 32], Dorvillé & Froehlich [29], Bispo & Froehlich [30], and Avelino-Capistrano *et al.* [33–35]. Specimens whose penial armature morphology did not correspond to that of any species previously described in the literature were considered new hypothetical species and included in the analyzes as different nominal species. After dissection, photos were made using a lucida camera mounted on a Leica DM1000 microscope and rendered using Adobe Illustrator CS6® editor. To capture and treat the pictures, a digital camera on a Leica M205A stereomicroscope and Adobe Photoshop CS3® editor were used, respectively. The geographic coordinates for constructing the occurrence maps for each species were compiled based on the information present in our examined material and previous records in the literature. Points of occurrence without coordinates provided by the authors were inferred. The maps were made using SimpleMappr resources [40], optimized in the QGIS Bucur 3.14.15 software (QGIS Development Team, 2020), and finished in Adobe Photoshop CS3® editor. The Brazilian state codes refer to ISO 3166–2:BR.

Total DNA was extracted from specimens using the DNeasy® Blood and Tissue Kit (Qiagen) following the manufacturer's protocol. Two legs from each specimen were used for extraction and a voucher number (S1 Table) was provided to each specimen and respective sequence. The primers LCO-1490 (5′-GGTCAACAAATCATAAAGATATTGG-3′) and HCO-2198 (5′-TAAACTTCAGGGTGACCAAAAAATCA-3′) [41] were employed to amplify the barcode region of the mitochondrial gene cytochrome c oxidase subunit I (COI) [42, 43] through polymerase chain reaction (PCR). The PCR program consisted of an initial denaturation step at 94°C for 5 minutes, followed by 5 cycles of denaturation at 94°C for 1 minute, annealing at 45°C for 1 minute, and extension at 72°C for 1 minute and 30 seconds, other 35

cycles of annealing at 50˚C for 1 minute, and a final extension step at 72˚C for 5 minutes. The amplicons were purified according to ExoSAP-IT™ factory protocol. The bidirectional sequencing was performed by the Center for Biological Resources and Genomic Biology (CRE-BIO/UNESP, Jaboticabal). All COI sequences used in this study are available in GenBank (S1 Table).

The editing of the chromatograms and the obtainment of consensus sequences were carried out in MEGA 11 software [44]. The consensus sequences were aligned using ClustalW [45], whilst the molecular distance matrix was inferred using Kimura-2-parameter (K2P; *d: Transitions + Transversions*; 1000 Bootstrap replicates; Uniform Rates; Pairwise deletion) in MEGA 11. A 503-base pair (bp) alignment of 48 sequences from the COI barcode region of 28 species of four genera was obtained. As outgroups, two species of *Anacroneuria* Klapálek 1909 [8], two of *Enderleina* Jewett 1960 [9], two of *Macrogynoplax* Enderlein 1909 [11], and one of *Paragripopteryx* Enderlein 1909 [11] were considered (S1 Table). Phylogenetic analyses were performed using Bayesian Inference and Maximum-Likelihood methods. For the Bayesian analysis, the best evolutionary models were chosen on Partitionfinder 2.1.1 software [46], being SYM+I+G, GTR+I, and GTR+G for the first, second, and third positions of the codon, respectively. Bayesian inference was carried out using MrBayes 3.2.2 [47] (2 independent runs of 4 Monte Carlo-Markov Chains for 2000000 generations, 25% generation burn-in), while maximum-likelihood analysis was performed using RAxML software [48] with 1000 searches for the best tree, including the GTR+G evolution model and a bootstrap with 5000 replications.

Species delimitation based on molecular data was assessed by the following methods: Automatic Barcode Gap Discovery (ABGD) considering primary (ABGD$_p$) and recursive (ABGD$_r$) partitions [49]; Assemble Species by Automatic Partitioning (ASAP) [50]; Poisson Tree Processes (PTP); and Bayesian implementation of the Poisson Tree Processes (bPTP) [51]. These analyses were carried out through online servers (i.e., ABGD, https://bioinfo.mnhn.fr/abi/public/abgd/; ASAP, https://bioinfo.mnhn.fr/abi/public/asap/; and PTP/bPTP, http://species.h-its.org/ptp/). For the ABGD and ASAP analyses, Kimura (K80) TS/TV with default parameters and a relative gap width of 1.0 (only for ABGD) was adopted, while for the PTP models, the trees resulting from maximum-likelihood and Bayesian analysis were used, together with the default settings.

### Nomenclatural acts

The electronic edition of this article conforms to the requirements of the amended International Code of Zoological Nomenclature, and hence the new names contained herein are available under that Code from the electronic edition of this article. This published work and the nomenclatural acts it contains have been registered in ZooBank, the online registration system for the ICZN. The ZooBank LSIDs (Life Science Identifiers) can be resolved and the associated information viewed through any standard web browser by appending the LSID to the prefix "http://zoobank.org/". The LSID for this publication is: urn:lsid:zoobank.org:pub: urn:lsid:zoobank.org:pub:E085D705-D1D0-434F-8F7A-C13C2BED03A9. The electronic edition of this work was published in a journal with an ISSN, and has been archived and is available from the following digital repositories: PubMed Central, LOCKSS.

## Results

### Molecular

Based on the COI fragment, the monophyly of the genus *Kempnyia* was recovered congruently in the Bayesian (BI) and maximum-likelihood (ML) analyses (Fig 1). The results of the species

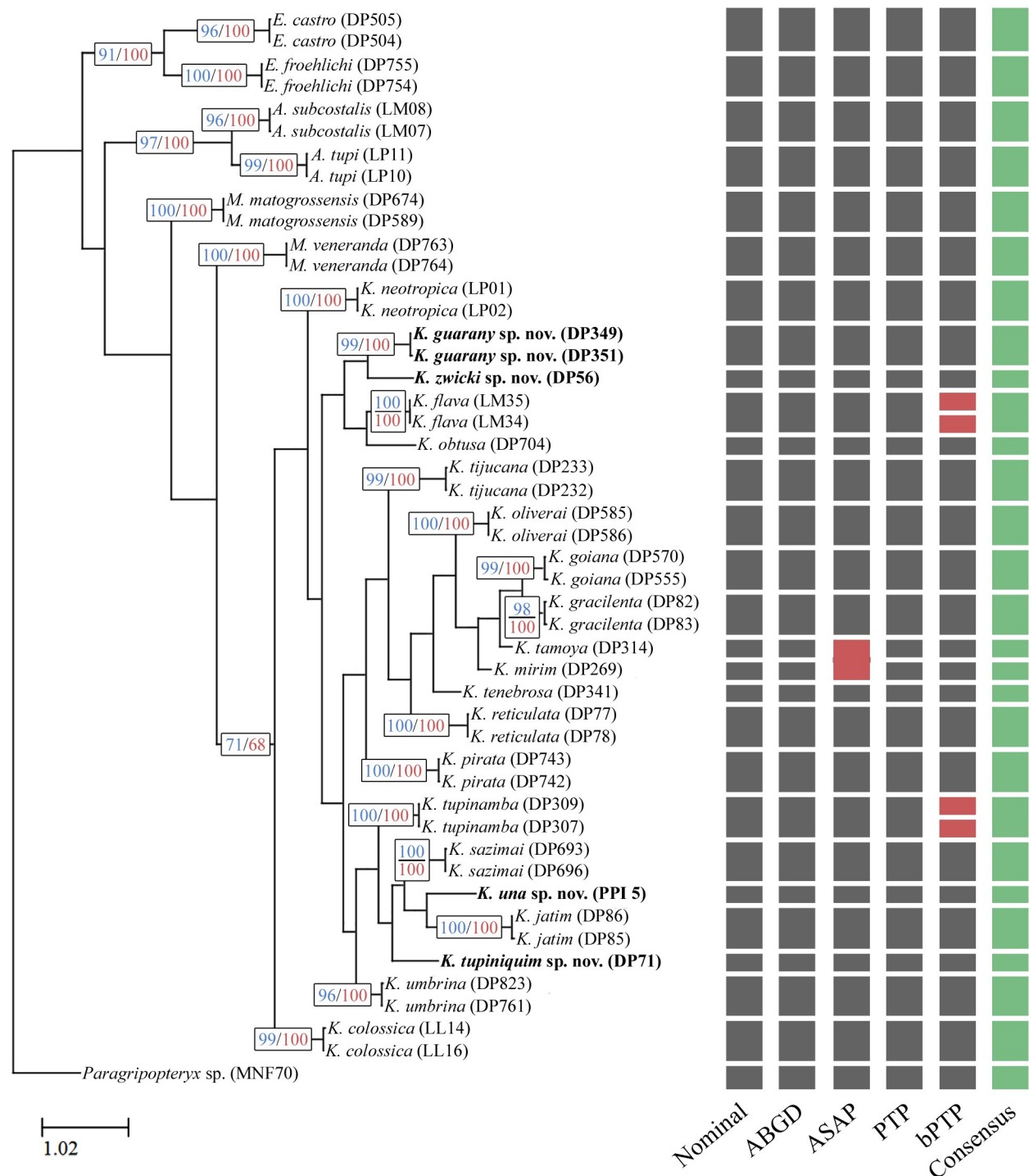

**Fig 1. Molecular analysis.** Maximum-likelihood tree of *Kempnyia* obtained using mitochondrial cytochrome c oxidase subunit I (COI) sequences (503 bp). Values are ML (blue) and BI (red) posterior probabilities. Vertical bars refer to results of different species delimitation analysis.

delimitation methods (ABGD, ASAP, PTP and bPTP) showed that the clusters were mostly in accordance with the previous morphological identification. The species delimitation methods that presented some incongruence when compared to the nominal morphological identification were ASAP and bPTP. Whereas ASAP grouped the species *Kempnyia tamoya* (DP314) and *K. mirim* (DP269), indicating that they form a single species with a molecular divergence of 8.1% and bPTP suggested that *Kempnyia flava* and *K. tupinamba* are not cohesive and should be divided into more species. In these cases, the molecular divergence was 0% between the sequences of *Kempnyia flava* and between *K. tupinamba*, evidencing some type of collapse or incapacity of the method adopted.

Therefore, the consensus between our molecular analyses corroborated with high clade support the morphological identification of the 21 species of *Kempnyia* in our sample, also showing a well-defined barcoding gap. The intraspecific divergence ranged from 0% between *Kempnyia flava* (LM34) and (LM35) to 1.4% between *K. gracilenta* (DP82) and (DP83), whilst the interspecific divergence varied from 8.1% between *Kempnyia mirim* (DP269) and *K. tamoya* (DP314) to 25.6% between *K. pirata* (DP742) and *K. una* **sp. nov**. (PPI 5). Our results supported the description of four new species: *Kempnyia guarany* **sp. nov**., *K. tupiniquim* **sp. nov**., and *K. zwicki* **sp. nov**. (male-based), and *K. una* **sp. nov**. (female-based).

## Morphological taxonomy

### *Kempnyia* **Klapálek, 1914**

#### *Kempnyia flava* **Klapálek, 1916** (Fig 2)

*Kempnyia flava* Klapálek, 1916: 53, 72, description [12]; Jewett, 1960: 176, illustration [9]; Illies, 1966: 340, catalog [19]; Zwick, 1972: 1167, illustration [25]; Zwick, 1973b: 276, catalog [52]; Froehlich, 1988: 153, illustration [21]; Stark, 2001: 415, checklist [22]; Bispo & Froehlich, 2004b: 109, record [53]; Stark *et al*., 2009: 124, checklist [13]; Nessimian *et al*., 2009: 316, record [54]; Froehlich, 2010: 180, catalog [36]; Froehlich, 2011a: 03, checklist [55]; Froehlich, 2011c: 21, record [31]; Gonçalves *et al*., 2017: 147, record [56]; Gonçalves *et al*., 2019: 105, checklist [57]; Almeida & Bispo, 2020: 19, COI sequence [23].

**Material examined. BR, ES**: Santa Teresa, i.2013, Salles & Lecci col., 1 male; REBIO Augusto Ruschi, Cachoeira da Estrada, 11.i.2016, Salles *et al*. col., 1 male; 20-21.ii.2018, Salles *et al*. col., 1 male; Córrego Bragacho, 21-22.x.2017, Salles *et al*. col., 1 male. **MG**: Ouro Preto, Vale do Tropeiro, Cachoeira do Abacaxi, 20˚12.270'S, 43˚38.163'W, 1120 m, 7.xi.2001, Paprocki col., 2 males and 2 females. Alto Caparaó, Parque Nacional do Caparaó, Rio Caparaó at Vale Verde, 20˚25.029'S, 41˚50.767'W, 1350 m, 12-13.iii.2002, Paprocki col., 1 male. **SP**: São Luiz do Paraitinga, Parque Estadual da Serra do Mar, Núcleo Santa Virgínia, Ribeirão Barro Branco, 20.i.2006, Bispo *et al*. col., 2 males. Iporanga, Parque Estadual Intervales, Rio do Carmo, bridge, 24˚18'15"S, 48˚24'31"W, 28.x.1999, 1 male; 20.ii.2000, 1 male. São Miguel Arcanjo, Parque Estadual Carlos Botelho, Rio Bonito, bridge, 24˚08'31.6"S, 47˚59'41.3"W, 06.ii.2017, Almeida *et al*. col., 2 females. Jundiaí, REBIO Serra do Japi, Córrego do Paraíso, 23˚14'29.3"S, 46˚57'17.6"W, 17.xi.2019, Almeida *et al*. col., 1 male and 1 female; 17.xi.2019, Almeida *et al*. col., 1 male; 01-06.xii.2019, Almeida *et al*. col., 1 male; Cachoeira das Bromélias, 23˚14'18.4"S, 46˚57'51.2"W, 17.xi.2019, Almeida *et al*. col., 3 males and 2 females.

**Measurement data**. Male (n = 14) forewing length: 13.5–16 mm (mean = 14.77 mm). Female (n = 8) forewing length: 19–20.5 mm (mean = 19.62 mm).

**Remarks**. The species is easily identified due to its characteristic penial armature [21] and general yellowish color. However, the specimens from Espírito Santo state have a darker pronotum than the others (Fig 2). This species was already recorded in Espírito Santo, Rio de Janeiro and São Paulo states [21, 31, 53, 54, 56] and now in Minas Gerais state (Fig 2).

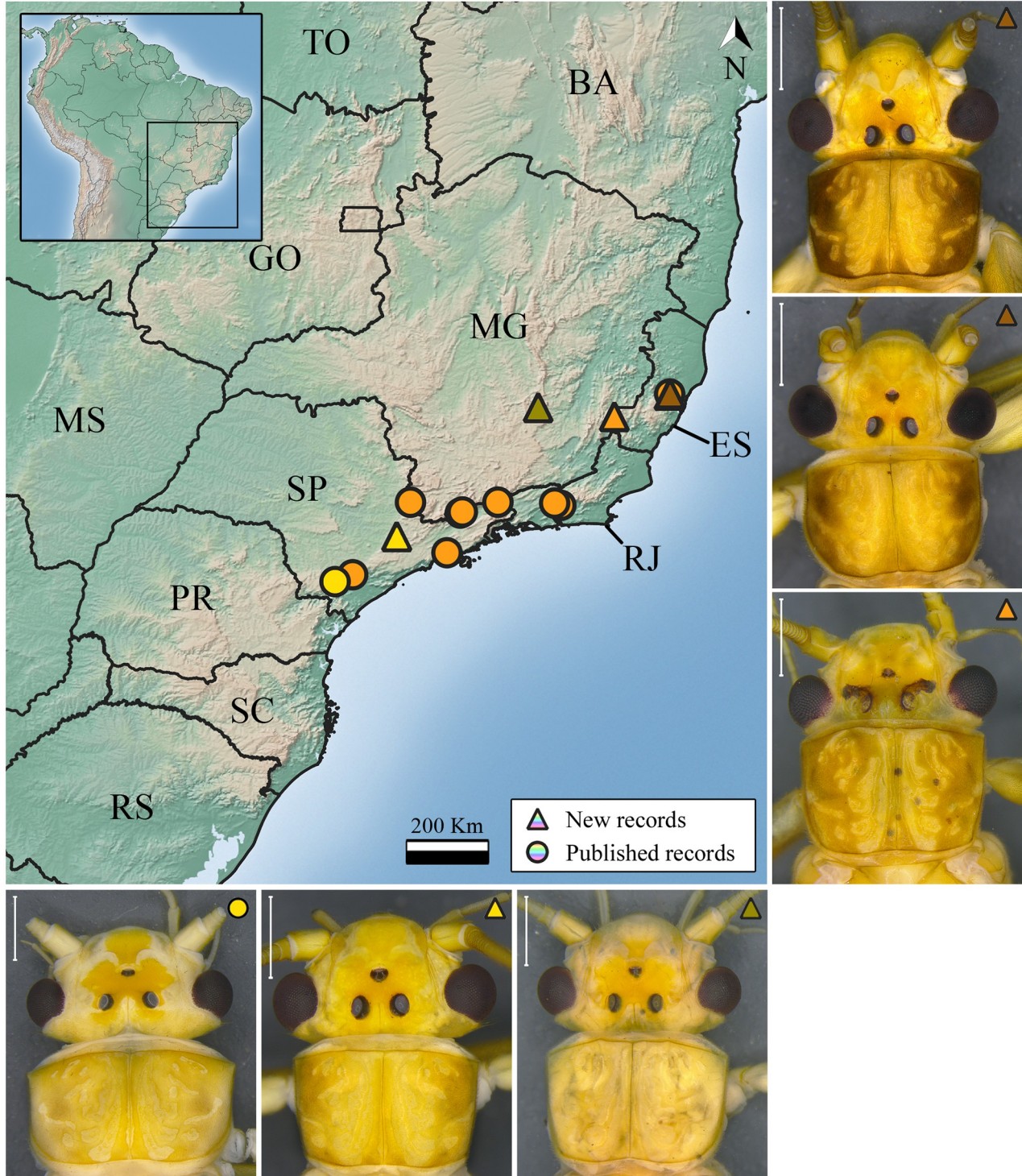

**Fig 2. Occurrence map and specimens of *Kempnyia flava*.** Scale: 1 mm. Brazilian states acronyms: Bahia—BA; Espírito Santo—ES; Goiás—GO; Mato Grosso—MT; Mato Grosso do Sul—MS; Minas Gerais—MG; Paraná—PR; Rondônia—RO; Rio de Janeiro—RJ; Rio Grande do Sul—RS; Santa Catarina—SC; São Paulo—SP; Tocantins—TO.

***Kempnyia goiana* Bispo and Froehlich, 2004** (Fig 3)

*Kempnyia goiana* Bispo and Froehlich, 2004a: 05, description [30]; Stark *et al.*, 2009: 124, checklist [13]; Froehlich, 2010: 180, catalog [36]; Rippel *et al.*, 2019b: 474, record [24].

**Material examined. BR, GO**: Pirenópolis, Parque Estadual dos Pireneus, Córrego Inferno, 21.xi.2021, Almeida & Taniguti col., 1 male. Alto Paraíso de Goiás, PARNA Chapada dos Veadeiros, Pouso Alto, 25.xi.2021, Almeida & Taniguti col., 1 male.

**Morphometric data**. Male (n = 2) forewing length: 11–12 mm (mean = 11.5 mm).

**Remarks**. *Kempnyia goiana* was first recorded in the municipalities of Pirenópolis and Jataí, in Goiás state [30], and later in Tocantins state [24]. Herein, we showed a new geographic record of the species (Fig 3).

***Kempnyia gracilenta* (Enderlein, 1909)** (Fig 4)

*Acroneuria gracilenta* Enderlein, 1909: 397, description [11]; *Eutactophlebia gracilenta* Klapálek, 1916: 67, new combination [12]; Jewett, 1960: 175, illustrations [9]; Illies, 1966: 332, catalog [19]; Zwick, 1973a: 490, illustrations [26]; *Kempnyia gracilenta* Zwick, 1983: 179, new combination [20]; Froehlich, 1984: 137, illustration [27]; Stark, 2001: 415, checklist [22]; Stark *et al.*, 2009: 124, checklist [13]; Froehlich, 2010: 180, catalog [36]; Froehlich, 2011a: 03, checklist [55]; Avelino-Capistrano *et al.*, 2011: 143, nymph and complementary description [58]; Avelino-Capistrano & Nessimian, 2014: 11, checklist [59]; Avelino-Capistrano *et al.*, 2014: 330, record [34]; Duarte *et al.*, 2014: 89, record [60]; Gonçalves *et al.*, 2017: 147, record [56]; Gonçalves *et al.*, 2019: 105, checklist [57].

**Material examined. BR, ES**: Santa Teresa, REBIO Augusto Ruschi, Córrego da Estrada, 27-28.xii.2017, Salles *et al.* col., 1 male; Córrego Bragacho, 18.xii.2017-17.i.2018, Salles *et al.* col., 2 males; 20-21.ii.2018, Salles *et al.* col., 1 male. **RJ**: Itatiaia, Parque Nacional do Itatiaia, Sítio da Acácias, 22º26.315'S, 44º36.625'W, 1300 m, 23-24.xi.2001, Paprocki col., 1 male.

**Measurement data**. Male (n = 5) forewing length: 13–14.1 mm (mean = 13.32 mm).

**Remarks**. The specimen from Itatiaia (Rio de Janeiro state) has a lighter central longitudinal stripe on the pronotum, while those from Espírito Santo state have a homogeneously pigmented pronotum (Fig 4). The latter also presented a larger penial armature than the former. However, the observed variations are normal within the species of this genus [23]. Herein, we showed new geographic records of the species (Fig 4).

***Kempnyia jatim* Froehlich, 1988** (Fig 4)

*Kempnyia jatim* Froehlich, 1988: 169, description [21]; Stark, 2001: 415, checklist [22]; Stark *et al.*, 2009: 124, checklist [13]; Froehlich, 2010: 181, catalog [36]; Froehlich, 2011a: 03, checklist [55]; Froehlich, 2011c: 22, record [32]; Duarte *et al.*, 2014: 89, record [60]; Gonçalves *et al.*, 2017: 147, record [56]; Gonçalves *et al.*, 2019: 105, checklist [57].

**Material examined. BR, BA**: Camacan, RPPN Serra Bonita, 2ª Cachoeira trilha, 04.xi.2009, Calor *et al.* col., 2 males. **ES**: Santa Teresa, REBIO Augusto Ruschi, Cachoeira da Estrada, 19.xi.2015, Salles *et al.* col., 5 males; Córrego da Estrada, 30.xi-01.x.2017, Salles *et al.* col., 1 male; 27-28.xii.2017, Salles *et al.* col., 1 male; 17-18.i.2018, Salles *et al.* col., 1 male; 20-21.iii.2018, Salles *et al.* col., 1 male. **MG**: Ouro Preto, Vale do Tropeiro, Cachoeira do Abacaxi, 7.xi.2001, Paprocki col., 4 males.

**Measurement data**. Male (n = 15) forewing length: 9.5–13.5 mm (mean = 10.51 mm).

**Remarks**. Although almost all the studied specimens are teneral, the shape of penial armature is in agreement with that of the holotype [21]. The color variations observed can be explained by the fact that the specimens from Espírito Santo state were teneral, while the others were preserved in 80% ethanol (Fig 4), which made them lose their pigmentation over time [23]. This species was already recorded in Bahia, Espírito Santo, Rio de Janeiro, and São Paulo states [21, 54, 56, 60] and now in Minas Gerais state (Fig 4).

***Kempnyia mirim* Froehlich, 1984** (Fig 3)

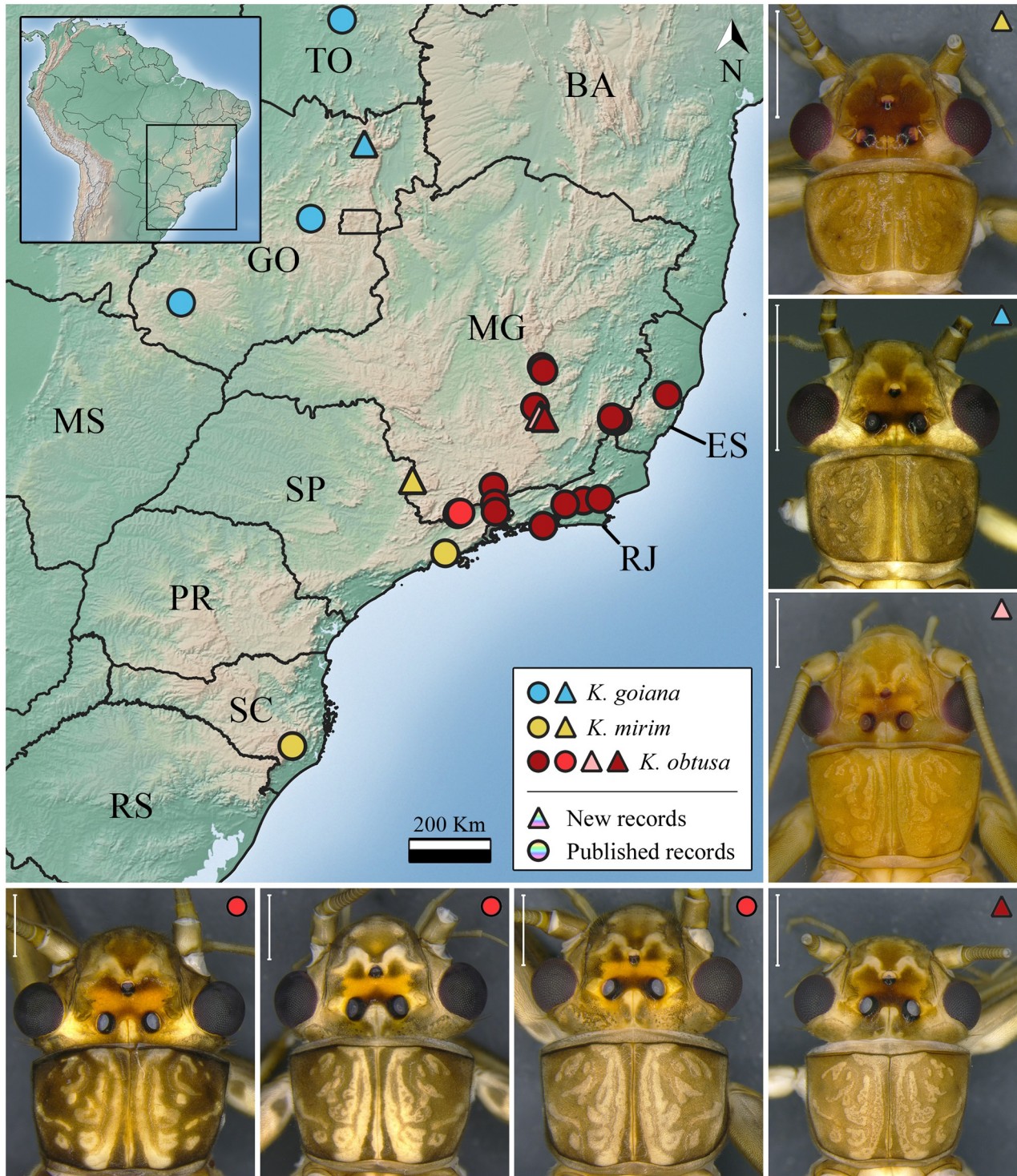

**Fig 3. Occurrence map and specimens of *Kempnyia goiana*, *K. mirim*, and *K. obtusa*.** Scale: 1 mm.

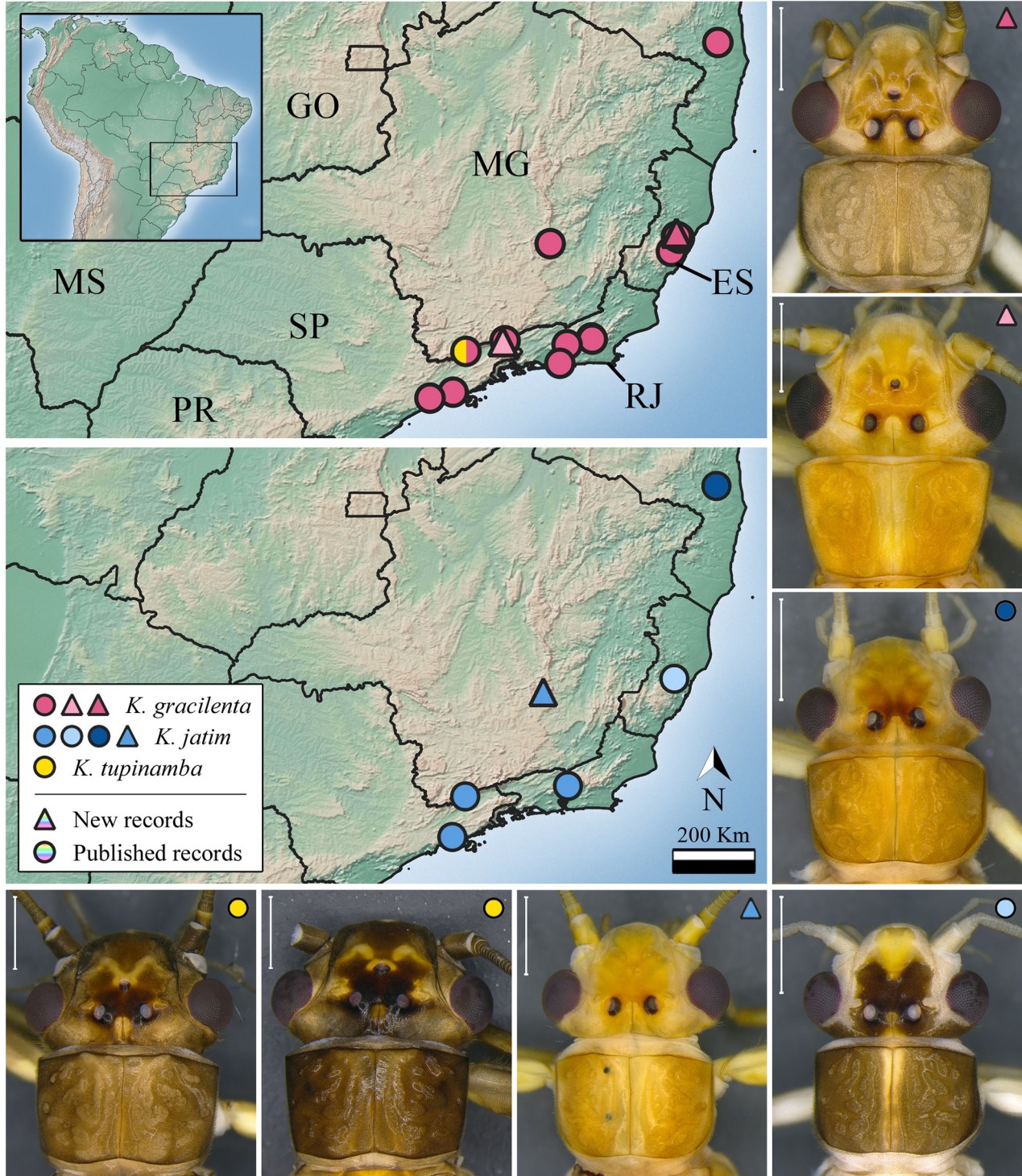

**Fig 4. Occurrence map and specimens of *Kempnyia gracilenta*, *K. jatim*, and *K. tupinamba*.** Scale: 1 mm. Light pink triangle: Specimens from Itatiaia, Rio de Janeiro state.

*Kempnyia mirim* Froehlich, 1984: 140, description [27]; Stark, 2001: 415, checklist [22]; Stark *et al.*, 2009: 124, checklist [13]; Froehlich, 2010: 181, catalog [36]; Froehlich, 2011a: 03, checklist [55]; Novaes & Bispo, 2014b: 283, record [61].

**Material examined. BR, MG**: Poços de Caldas, Morro do Ferro, Fazenda de Eucalipto, 21˚53'31.68"S, 46˚32'57"W, 08.iii.2019, Fusari *et al.* col., 20 males and 4 females.

**Measurement data**. Male (n = 20) forewing length: 9–10.1 mm (mean = 9.67 mm). Female (n = 4) forewing length: 11–12.1 mm (mean = 11.7 mm).

**Remarks**. *Kempnyia mirim* is one of the smallest species of *Kempnyia*. It is similar to *K. pinhoi* Froehlich, 2011 [32], mainly in the size and shape of penial armature. When describing *K. pinhoi* [32] identified some differences in the general color and penial armature, which made him separate both species. Nonetheless, in this study we verified some differences in the color pattern between teneral and non-teneral specimens, in addition to the effects on color caused by preservation in 80% ethanol [23]. Thus, it is possible that both species are actually the same. For this reason, they should be restudied. *Kempnyia mirim* was already recorded in São Paulo and Santa Catarina states [27, 61] and now in Minas Gerais state (Fig 3).

### *Kempnyia neotropica* (Jacobson & Bianchi, 1905) (Fig 5)

*Perla* (*Perla*) *obscura* Pictet, 1841: 28, description [62]; *Perla neotropica* Jacobson & Bianchi, 1905: 617, new name [63]; *Macrogynoplax aterrima* Klapálek, 1916: 73, description [12]; *Kempnyia neotropica*, Zwick, 1972: 1168, new combination and illustrations [25]; Zwick, 1973a: 276, record [26]; Bispo & Froehlich, 2004a: 2, record [30]; Bispo & Froehlich, 2004b: 107, record [53]; Bispo & Froehlich, 2008; 62, nymph and complementary description [64]; Stark, 2001: 415, checklist [22]; Stark *et al.*, 2009: 124, checklist [13]; Froehlich, 2010: 181, catalog [36]; Froehlich, 2011b: 133, illustration [32]; Froehlich, 2011c: 22, record [31]; Duarte *et al.*, 2014: 89, record [60]; Novaes & Bispo, 2014a: 464, illustration and picture [65]; Novaes & Bispo, 2014b: 283, illustration and picture [61]; Novaes & Bispo, 2014c: 439, record [66]; Novaes *et al.*, 2016: 98, record [67]; Gonçalves *et al.*, 2017: 147, record [56]; Gonçalves *et al.*, 2019: 107, checklist [57]; Almeida & Bispo, 2020: 19, COI sequence [23].

*Kempnyia petersorum* Froehlich, 1996: 119, description [28]; Froehlich, 2011b: 134, illustration [32].

**Material examined. BR, BA**: Wenceslau Guimarães, Estação Ecológica de Wenceslau Guimarães, Riacho Semo Grande, Cachoeira em cima, 10.x.2010, Calor *et al.* col., 1 male. **ES**: Santa Teresa, REBIO Augusto Ruschi, Cachoeirinha da Estrada, 19.xi.2015, Salles *et al.* col., 1 male; Cachoeirinha à Montante, 19.xi.2015, Salles *et al.* col., 1 male. Parque Nacional do Caparaó, Santa Marta, Rio Santa Marta, Sede, 18-19.ii.2016, Salles *et al.* col., 3 males. **MG**: Ouro Preto, Vale do Tropeiro, Cachoeira do Abacaxi, 07.xi.2001, Paprocki col., 3 males. Alto Caparaó, Rio Caparaó, Hotel Parque Caparaó, 20º25.498'S, 41º51.500'W, 830 m, 11-14.iii.2002, Paprocki col., 1 male. Poços de Caldas, Fazenda de Eucalipto, 08.iii.2019, Fusari *et al.* col., 1 male. **RJ**: Itatiaia, Parque Nacional do Itatiaia, Córrego Campo Belo, 798 m, 06.ix.2017, 7 males; 08.x.2017, Dias & Campos col., 1 male. **SP**: Iporanga, Parque Estadual Intervales, 7.ii.1989, 1 male (*K. petersorum* Holotype/MZUSP); Parque Estadual Intervales, Rio do Carmo, 27.ii.1997, Melo col., 1 male; 14.xii.2014, Bispo col., 7 females; 08.ii.2017, Almeida *et al.* col., 1 male and; 09.ii.2017, Almeida *et al.* col., 3 males; Ribeirão Água Comprida, 24˚17'38"S, 48˚25'04"W, 29.xi.2000, 1 male. São Miguel Arcanjo, Parque Estadual Carlos Botelho, Ribeirão de Pedras, bridge, 24˚03'40"S, 47˚59'51"W, 06.ii.2017, Almeida *et al.* col., 2 males. Campos do Jordão, Parque Estadual Campos do Jordão, Córrego Serrote, 22˚39'30"S, 45˚26'32"W, 18.xii.2018-15.i.2019, Almeida *et al.* col., 1 female; Córrego Galharada, 22˚41'29"S, 45˚27'58"W, 13.ii.2019, Almeida *et al.* col., 1 female; 08.xii.2019, Almeida *et al.* col., 1 male. Jundiaí, REBIO Serra do Japi, Córrego do Paraíso, 15-20.xi.2019, Almeida *et al.* col., 3 male; 17.xi.2019, Almeida *et al.* col., 2 males; 20.xii.2019-07.i.2020, Almeida *et al.* col., 1 male.

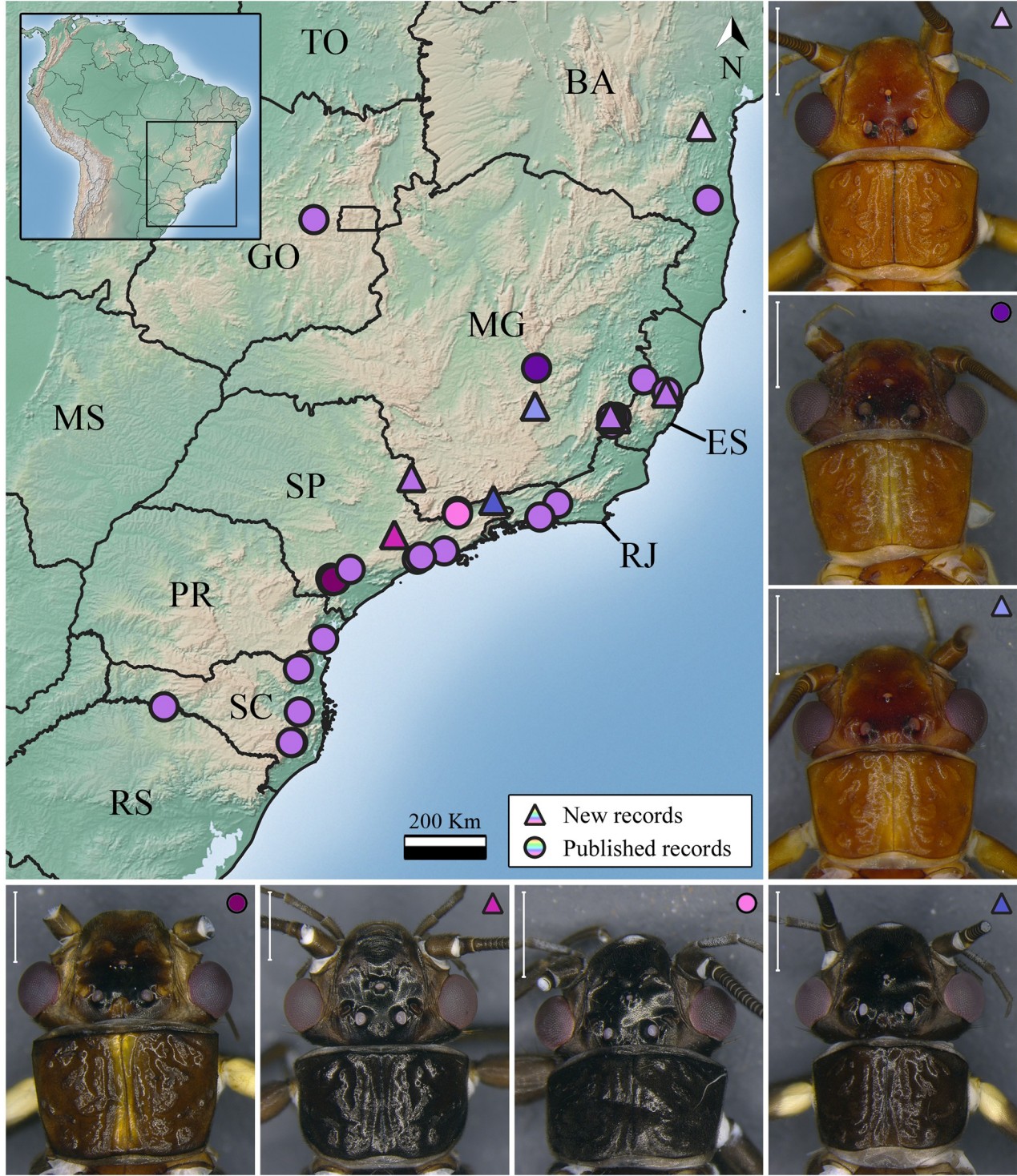

**Fig 5. Occurrence map and specimens of *Kempnyia neotropica*.** Scale: 1 mm.

**Measurement data**. Male (n = 33) forewing length: 11.5–14.2 mm (mean = 12.55 mm). Female (n = 9) forewing length: 16–17.5 mm (mean = 16.83 mm).

**Remarks**. Almeida & Bispo [23] observed important variations in the pattern of spots on the head and pronotum of *K. neotropica*. The studied specimens presented all of these variations (Fig 4). The penial armature also revealed some variation, mainly with respect to the hooks, which ranged from cylindrical to flat. Herein, we showed new geographic records of the species (Fig 5).

*Kempnyia obtusa* **Klapálek, 1916** (Fig 3)

*Kempnyia obtusa* Klapálek, 1916: 51, description [12]; Illies, 1966: 340, catalog [19]; Zwick, 1972: 1171, illustrations [25]; Froehlich, 1988: 153, illustrations [21]; Stark, 2001: 415, checklist [22]; Stark *et al.*, 2009: 124, checklist [13]; Froehlich, 2010: 181, catalog [36]; Froehlich, 2011a: 03, checklist [55]; Froehlich, 2011c: 22, illustration [31]; Avelino-Capistrano *et al.*, 2014: 330, nymph description [34]; Novaes & Bispo, 2014c: 439, record [66]; Gonçalves *et al.*, 2017: 148, record [56]; Gonçalves *et al.*, 2019: 107, checklist [57].

**Material examined. BR, MG**: Ouro Preto, Parque Estadual do Itacolomi, trib. to Rio Belchior, 20º25.302'S, 43º25.697'W, 700 m, 06.xi.2001, Paprocki col., 4 males. Estação Ecológica de Tripuí, Córrego Botafogo, 20º22.908'S, 43º33.615'W, 1100 m, 25.xi.2001, Paprocki col., 3 males; Rio dos Velhos, Cachoeira Catarina Mendes, 09.i.2017, 1 male. **SP**: Campos do Jordão, Parque Estadual Campos do Jordão, Córrego Serrote, 18.xii.2018-15.i.2019, Fusari *et al.* col., 2 males; Córrego Galharada, 05.xii.2019, Almeida *et al.* col., 4 males; 07.xii.2019, Almeida *et al.* col., 3 males; 08.xii.2019, Almeida *et al.* col., 1 male; 09.xii.2019, Almeida *et al.* col., 3 males.

**Measurement data**. Male (n = 21) forewing length: 15.1–20 mm (mean = 17.27 mm).

**Remarks**. The species can be easily identified since it has a characteristic penial armature [25]. The nymph is one of the most easily identified due to the unique pattern of spots on the head and pronotum [34]. We observed consistency in the pattern of spots on the head and pronotum of the studied specimens, with some variation in the shades of the spots (Fig 3). Herein, we showed new geographic records of the species (Fig 3).

*Kempnyia pirata* **Froehlich, 2011** (Fig 6)

*Kempnyia pirata* Froehlich, 2011c: 23, description [31].

**Material examined. BR, ES**: Santa Teresa, REBIO Augusto Ruschi, Córrego à Montante, 09.xii.2015, Salles *et al.* col., 1 female; Córrego Bragacho, 28.iv-27.v.20117, Salles *et al.* col., 1 female; Córrego da Estrada, 18.xii.2017-17.i.2018, Salles *et al.* col., 1 female; Roda D'agua, 18.xii.2017-17.i.2018, Salles *et al.* col., 1 female. **RJ**: Teresópolis, PARNA Serra dos Órgãos, trilha da Pedra do Sino, Cachoeira do Papel, 30.xi.2021-23.i.2022, 22˚27'08.5"S 43˚00'54.8"W, 1700m, malaise, Vaz & Alves col., 3 females. **SP**: Jundiaí, REBIO Serra do Japi, Córrego do Paraíso, 20.xii.2019-07.i.2020, Almeida *et al.* col., 1 male.

**Measurement data**. Male (n = 1) forewing length: 8 mm, Female (n = 7) forewing length: 10–11.8 mm (mean = 10.87 mm).

**Remarks**. The species is easily identified due to its characteristic penial armature, black general color, and wings with a colorless window [31]. It was already recorded in the Mantiqueira Mountains, São Paulo state [31], and now in another location in São Paulo state and in Espírito Santo and Rio de Janeiro states (Fig 6).

*Kempnyia reichardti* **Froehlich, 1984** (Fig 7)

*Kempnyia reichardti* Froehlich, 1984: 143, description [27]; Stark, 2001: 415, checklist [22]; Stark *et al.*, 2009: 124, checklist [13]; Froehlich, 2010: 182, catalog [36]; Froehlich, 2011a: 03, checklist [55]; Froehlich, 2011c: 26, illustration [31].

**Material examined. BR, MG**: Parque Estadual do Ibitipoca, spring trib. Near director's house, 21˚42.695'S, 43˚53.760'W, 1357 m, 19-20.xi.2001, Paprocki & Blahnik col., 1 male. **SP**: Campos do Jordão, Parque Estadual Campos do Jordão, Córrego Galharada, 12-13.ii.2019,

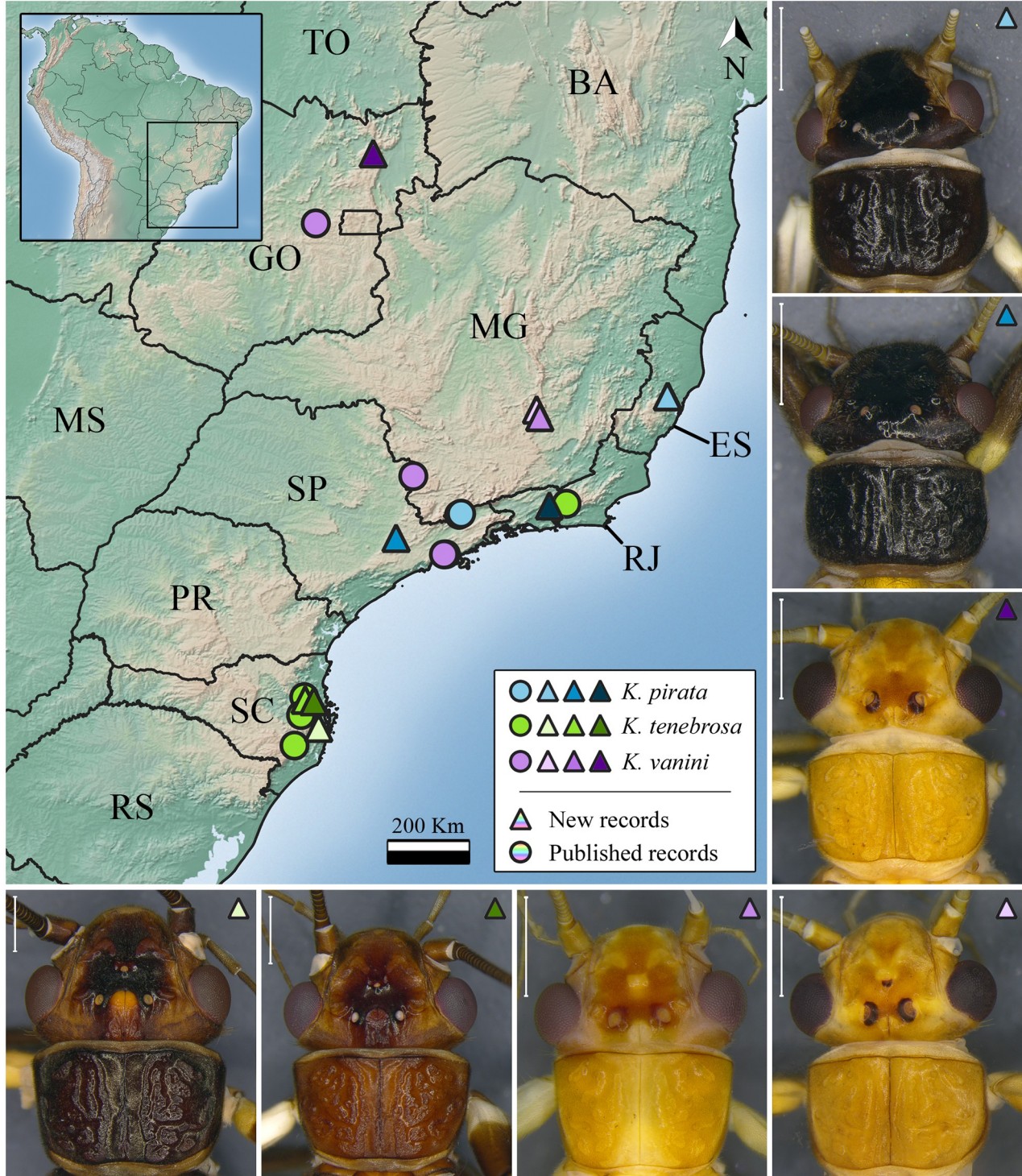

**Fig 6. Occurrence map and specimens of *Kempnyia pirata*, *K. tenebrosa*, and *K. vanini*.** Scale: 1 mm. Light blue circle: Mantiqueira Mountains. Light green triangle: Specimen from Tabuleiro Mountains; Dark green triangle: Specimen from Brusque.

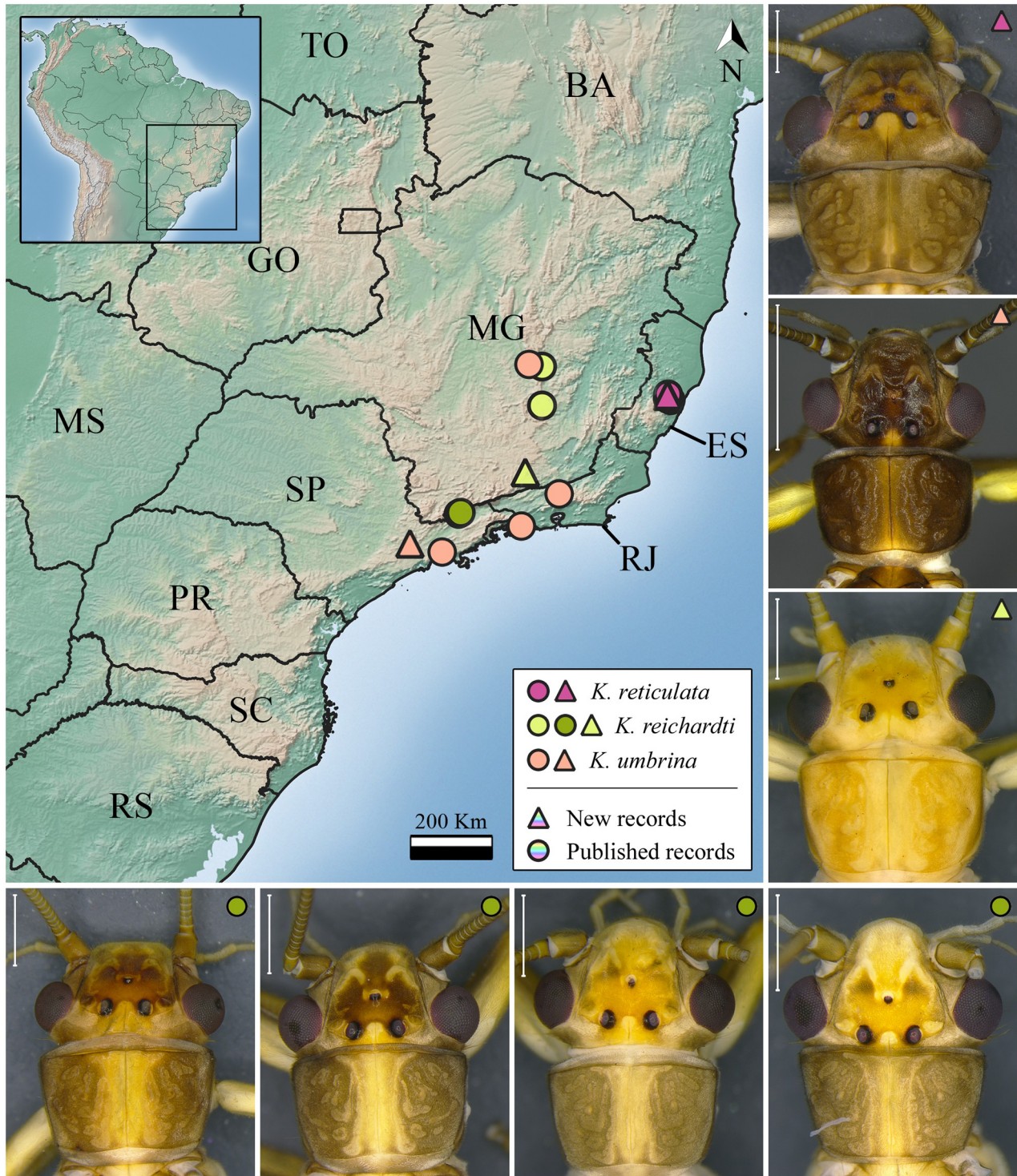

**Fig 7. Occurrence map and specimens of *Kempnyia reichardti*, *K. reticulata*, and *K. umbrina*.** Scale 1 mm.

Almeida *et al.* col., 1 male; 14.ii.2019, Almeida *et al.* col., 1 male; 05.xii.2019, Almeida *et al.* col., 1 male; 07.xii.2019, Almeida *et al.* col., 1 male.

**Measurement data**. Male (n = 5) forewing length: 13.8–15 mm (mean = 14.36 mm).

**Remarks**. The studied specimens are in agreement with the holotype [27]. Nonetheless, we identified some variations in body size and pattern of spots on the head, as some specimens have brown or yellow spots on the frons (Fig 7). These variations approximate the limits between *Kempnyia reichardti* and *K. tamoya* Froehlich, 1984 [27]. Perhaps, a molecular approach could solve any possible questions that may arise during the identification of both species. Herein, we showed a new geographic record of the species (Fig 7).

*Kempnyia reticulata* **(Klapálek, 1916)** (Fig 7)

*Eutactophlebia reticulata* Klapálek, 1916: 46, description [12]; Illies, 1966: 333, catalog [19]; *Kempnyia reticulata* Zwick, 1983: 177, new combination and illustrations [20]; Stark, 2001: 415, checklist [22]; Stark *et al.*, 2009: 124, checklist [13]; Froehlich, 2010: 182, catalog [36]; Avelino-Capistrano *et al.*, 2011: 144, nymph and complementary description [58]; Avelino-Capistrano & Nessimian, 2014: 12, record [59]; Gonçalves *et al.*, 2019: 107, checklist [57].

**Material examined. BR, ES**: Santa Teresa, REBIO Augusto Ruschi, Córrego Bragacho, 21-22.vi.2017, Salles *et al.* col., 1 male; 21.vi-26.vii.2017, Salles *et al.* col., 1 male; 26-27.vii.2017, Salles *et al.* col., 1 male; 26vii-23.viii.2017, Salles *et al.* col., 1 male; Córrego entre estradas, 21.vi-26.vii.2017, Salles *et al.* col., 1 male; Roda D'agua, 26.vii-23.viii.2017, Salles *et al.* col., 2 males.

**Measurement data**. Male (n = 7) forewing length: 21–25.1 mm (mean = 23.14 mm).

**Remarks**. The species is easily identified due to its characteristic penial armature [20]. It was previously recorded only in Espírito Santo state [12, 20, 58], and herein we included three new geographic records (Fig 7).

*Kempnyia serrana* **(Navás, 1936)** (Fig 8)

*Diperla serrana* Navás, 1936: 729, description [15]; Aubert, 1956: 439, catalog [68]; Illies, 1966: 476, catalog [19]; *Eutactophlebia gracilenta* Zwick, 1973b: 20, illustration [26]; *Eutactophlebia serrana* Froehlich, 1979: 70, new combination [69]; *Kempnyia serrana* Zwick, 1983: 179, new combination and illustrations [20]; Froehlich, 1984: 139, complementary description [27]; Stark, 2001: 415, checklist [22]; Stark *et al.*, 2009: 124, checklist [13]; Froehlich, 2010: 182, catalog [36]; Froehlich, 2011a: 03, checklist [55]; Gonçalves *et al.*, 2019: 109, checklist [57].

**Material examined. BR, SP**: São Carlos, Fazenda Embrapa, Córrego Canchim, 21˚57'07"S, 47˚50'12"W, 15.ix.2007, Roque col., 2 males and 1 female.

**Measurement data**. Male (n = 2) forewing length: 12.9–13.5 mm (mean = 13.2 mm). Female (n = 1) forewing length: 14.9 mm.

**Remarks**. Herein, we expanded the species distribution to the countryside of São Paulo state (Fig 8).

*Kempnyia tamoya* **Froehlich, 1984** (Fig 8)

*Kempnyia tamoya* Froehlich, 1984: 145, description [27]; Stark, 2001: 415, checklist [22]; Stark *et al.*, 2009: 124, checklist [13]; Froehlich, 2010: 182, catalog [36]; Froehlich, 2011a: 03, checklist [55]; Froehlich, 2011c: 28, illustrations [31].

**Material examined. BR, SP**: Jundiaí, REBIO Serra do Japi, Cachoeira do Chá II, 23˚14'34.7"S, 46˚56'06.7"W, 16.xi.2019, Almeida *et al.* col., 2 males; Biquinha, 23˚14'22.2"S, 46˚56'07.2"W, 16.xi.2019, Almeida *et al.* col., 2 males; Córrego do Paraíso, 17.xi.2019, Almeida *et al.* col., 3 males; 17.xi.2019, Almeida *et al.* col., 3 males; 18.xi.2019, Almeida *et al.* col., 1 male; Cachoeira do Chá I, 23˚14'17.3"S, 46˚56'00.7"W, 18.xi.2019, Almeida *et al.* col., 4 males. Campos do Jordão, Parque Estadual Campos do Jodrão, Córrego Galharada, 05.xii.2019, Almeida *et al.* col., 4 males; 06.xii.2019, Almeida *et al.* col., 1 male; 07.xii.2019, Almeida *et al.* col., 5 males; 08.xii.2019, Almeida *et al.* col., 1 male; 09.xii.2019, Almeida *et al.* col., 2 males.

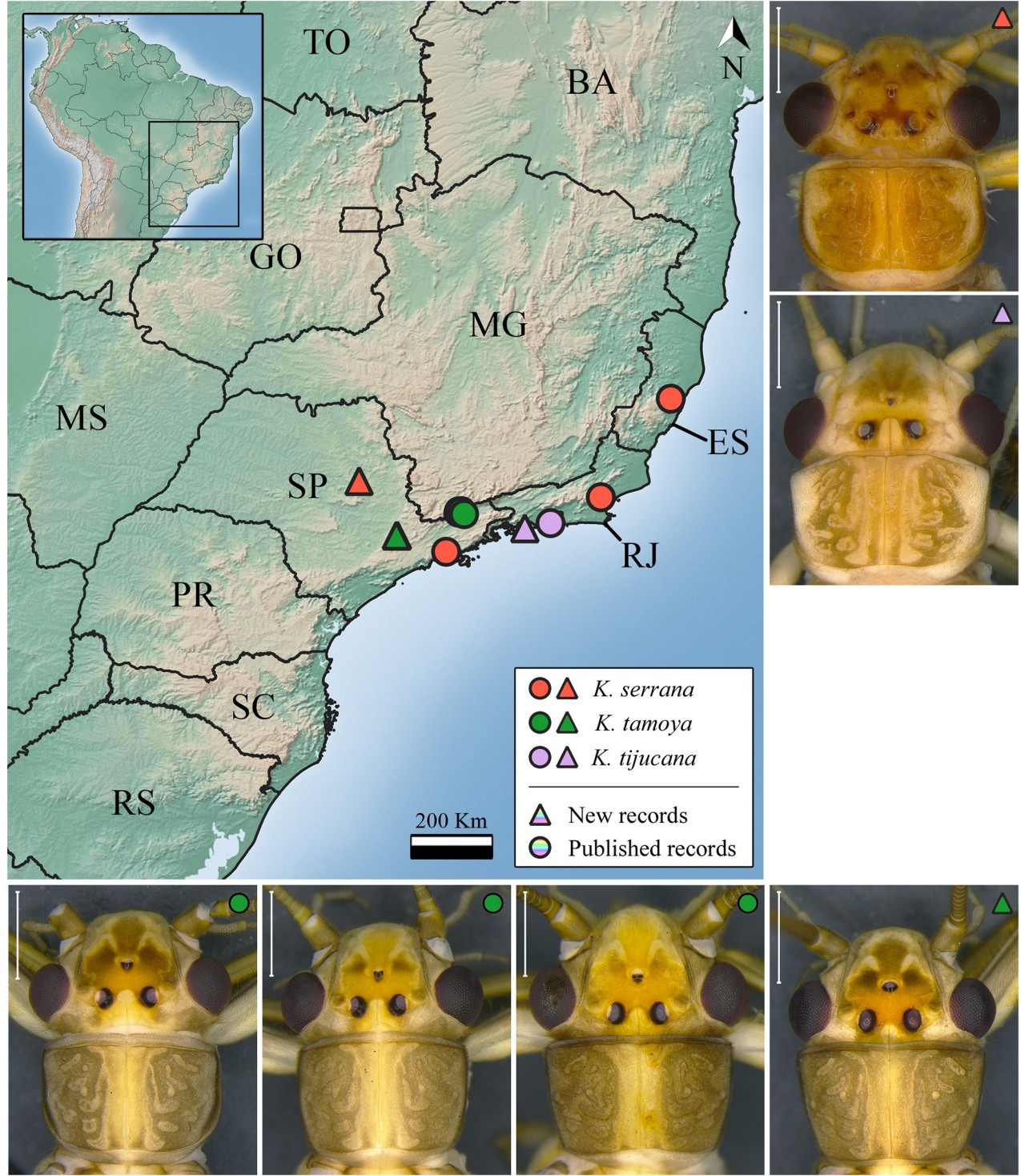

**Fig 8. Occurrence map and specimens of *Kempnyia serrana*, *K. tamoya*, and *K. tijucana*.** Scale: 1 mm.

**Measurement data**. Male (n = 29) forewing length: 12–15.5 mm (mean = 14.39 mm).

**Remarks**. The studied specimens are in agreement with the holotype [27]. However, there are specimens with wider penial armature in dorsal and ventral views and with darker spots on the head (Fig 8). Herein, we expanded the distribution of the species, previously recorded only for a few streams in a location in São Paulo state [27, 31], to the municipality of Jundiaí, São Paulo state (Fig 8).

*Kempnyia tenebrosa* **Klapálek, 1916** (Fig 6)

*Kempnyia tenebrosa* Klapálek, 1916: 50, description [12]; Jewett, 1960: 177, notes [9]; Illies, 1966: 340, catalog [19]; Zwick, 1972: 1172, illustrations [25]; Stark, 2001: 415, checklist [22]; Stark *et al.*, 2009: 124, checklist [13]; Froehlich, 2010: 182, catalog [36]; Froehlich, 2011b: 136, illustration [32]; Novaes & Bispo, 2014b: 283, record and illustrations [61].

**Material examined. BR, SC**: Santo Amaro Imperatriz, Serra do Tabuleiro, 13.ii-2.viii.2014, Pinho col., 1 male. Brusque, RPPN Chácara Edith, 21.vii.2018, Pinho col., 1 female; 04.v-3.vi.2019, Pinho col., 2 males; i-ii.2019, Pinho col., 3 male and 1 female. Blumenau, Parque Nacional da Serra do Itajaí, Parque das Nascentes, Trilha do Morro do Sapo, Córrego da Placa, 05.xi.2019, Almeida & Miguel col., 1 female.

**Measurement data**. Male (n = 6) forewing length: 12.2–17 mm (mean = 13.81 mm). Female (n = 3) forewing length: 15.5–16.5 mm (mean = 16.16 mm).

**Remarks**. The studied male specimens are similar to the holotype from Rio de Janeiro state [25]. Nevertheless, they showed some variations in body size, pattern of spots on the head and pronotum (Fig 6), and shape of penial armature. The specimen from the Tabuleiro Mountains (Santa Catarina state) is larger (17.5 mm) and has differences in the shape of penial armature, mainly in lateral view, compared to those from Brusque (Santa Catarina state) (12.2–13.5 mm). The penial armature of the specimen from the Tabuleiro Mountains is more similar to that illustrated by Novaes & Bispo [61], while those from Brusque are comparable to that illustrated by Froehlich [32]. A molecular approach would help better understand these variations. Herein, we showed new geographic records of the species (Fig 6).

*Kempnyia tijucana* **Dorvillé & Froehlich, 1997** (Fig 8)

*Kempnyia tijucana* Dorvillé & Froehlich, 1997: 178, description [29]; Dorvillé & Froehlich, 2001: 385, nymph description [70]; Stark, 2001: 415, checklist [22]; Stark *et al.*, 2009: 124, checklist [13]; Froehlich, 2010: 182, catalog [36].

**Material examined. BR, RJ**: Mangaratiba, Ilha da Marambaia, Rio Marambaia, 23˚04'12.1"S, 43˚55'06.8"W, 20.x-24.xi.2018, Avelino-Capistrano col., 1 female; 15.vii-18.viii.2018, Avelino-Capistrano col., 1 male.

**Measurement data**. Male (n = 1) forewing length: 12.1 mm, Female (n = 1) forewing length: 13 mm.

**Remarks**. The studied male specimen presented no asymmetrical arrangement of the penial armature, as illustrated by Dorvillé & Froehlich [29]. Herein, we showed a new geographic record of the species (Fig 8).

*Kempnyia tupinamba* **Froehlich, 2011** (Fig 4)

*Kempnyia tupinamba* Froehlich, 2011c: 28, description [31].

**Material examined. BR, SP**: Campos do Jordão, Parque Estadual Campos do Jordão, Córrego Galharada, 07.xii.2019, Almeida *et al.* col., 3 males.

**Measurement data**. Male (n = 3) forewing length: 14.2–16 mm (mean = 15.06 mm).

**Remarks**. One of the studied specimens presented a variation in the pattern of spots on the head, being slightly different from the description [31]. This specimen showed a fragmented M-line forming three yellow spots (Fig 4).

*Kempnyia umbrina* **Froehlich, 1988** (Fig 7)

*Kempnyia umbrina* Froehlich, 1988: 164, description [21]; Stark, 2001: 415, checklist [22]; Stark *et al.*, 2009: 124, checklist [13]; Froehlich, 2010: 182, catalog [36]; Froehlich, 2011a: 03, checklist [55].

**Material examined. BR, SP**: Jundiaí, REBIO Serra do Japi, Córrego do Paraíso, 17.xi-17. xii.2022, Almeida & Taniguti col., 2 males. Salesópolis, Estação Biológica de Boracéia, Córrego Venerando, 09.xii.2022, Almeida & Taniguti col., 1 female; Córrego Coruja, 22.i.2023, Almeida & Taniguti col., 1 female.

**Morphometric data**. Male (n = 2) forewing length: 9.6–9.8 mm (mean = 9.7 mm).

**Remarks**. This species was previously recorded in Minas Gerais, Rio de Janeiro and São Paulo states [21]. Herein, we expanded its distribution to the countryside of São Paulo state (Fig 7).

*Kempnyia vanini* **Froehlich, 1988** (Fig 6)

*Kempnyia vanini* Froehlich, 1988: 164, description [21]; Bispo & Froehlich, 2004a: 02, record [30]; Stark, 2001: 415, checklist [22]; Stark *et al.*, 2009: 124, checklist [13]; Froehlich, 2010: 182, catalog [36]; Froehlich, 2011a: 03, checklist [55].

**Material examined. BR, GO**: Alto Paraíso de Goiás, Chapada dos Veadeiros, Córrego Loquinhas, 15.xii.2006, Bispo col., 1 male. **MG**: Ouro Preto, Vale do Tropeiro, Cachoeira do Abacaxi, 7.xi.2001, Paprocki col., 3 males. Estação Ecológica de Tripuí, Córrego Botafogo, 25. xi.2001, Paprocki col., 2 males. **SP**: Salesópolis, Estação Biológica de Boracéia, 22.xii.1987, Froehlich col., 1 male.

**Measurement data**. Male (n = 7) forewing length: 10.8–13 mm (mean = 11.77 mm).

**Remarks**. The studied specimens showed a variation in the pattern of spots on the head and number of ocelli. The general color variation was likely due to preservation in 80% ethanol [23]. The number of ocelli ranged from two in populations from Goiás and Minas Gerais states to three in the population from São Paulo state and a population from Minas Gerais state (Tripuí Ecological Station) (Fig 6). What we probably know as *K. vanini* is, in reality, a species complex. The penial armature of *Kempnyia vanini* can be confused with that of species such as *K. tupinamba*, *K. sazimai* Froehlich, and *K. umbrina*. For all these reasons, we suggest that a molecular study including all populations of these species be carried out in order to better understand the limits among species. Despite this, we expanded the distribution of the species to the north of Goiás state and the countryside of Minas Gerais state (Fig 6).

*Kempnyia guarany* **sp. nov. Almeida, Gonçalves & Bispo** (Fig 9A–9G)

urn:lsid:zoobank.org:act:3D9142BC-16F2-41A6-BC91-B14E0B441EF1

**Material examined. BR, SC**: Blumenau, Parque Nacional da Serra do Itajaí, Parque das Nascentes, Encontro das águas, 03.xi.2019, Almeida & Miguel col., 1 male (Holotype/MZUSP), 1 male (Paratype/CLBA).

**Measurement data**. Male (n = 2) forewing length: 16–17 mm (mean = 16.5 mm).

**Description**. General color brown to yellowish. Anterior ocellus present. Frons brown, lighter along sides of frons and genae; M-line yellowish; lappets brown, frontoclypeus, in front of M-line, ochraceous; parietalia brown, yellow in the middle, near the coronal suture (Fig 9A). Antennae brown. Pronotum trapezoidal, brown with yellowish longitudinal and centralized stripe (Fig 9A). Membrane and veins of wings brown and dark brown, respectively (Fig 9B). Legs yellow to brown; coxae, trochanters, and base of femora yellow; apex of femora dark brown; tibia and tarsus dark brown. Cercomeres yellowish.

**Male**. Tergum X pale medially and at bases of cerci, most sensilla basiconica carrot-shaped. Projection of the subgenital plate about twice as broad as long; T-shaped hammer white, prominent in lateral view; from the base to the tip of the hammer there is a light ocher stripe; subgenital plate mostly whitish (Fig 9C and 9D). Paraprocts finger-like, without a subapical denticle;

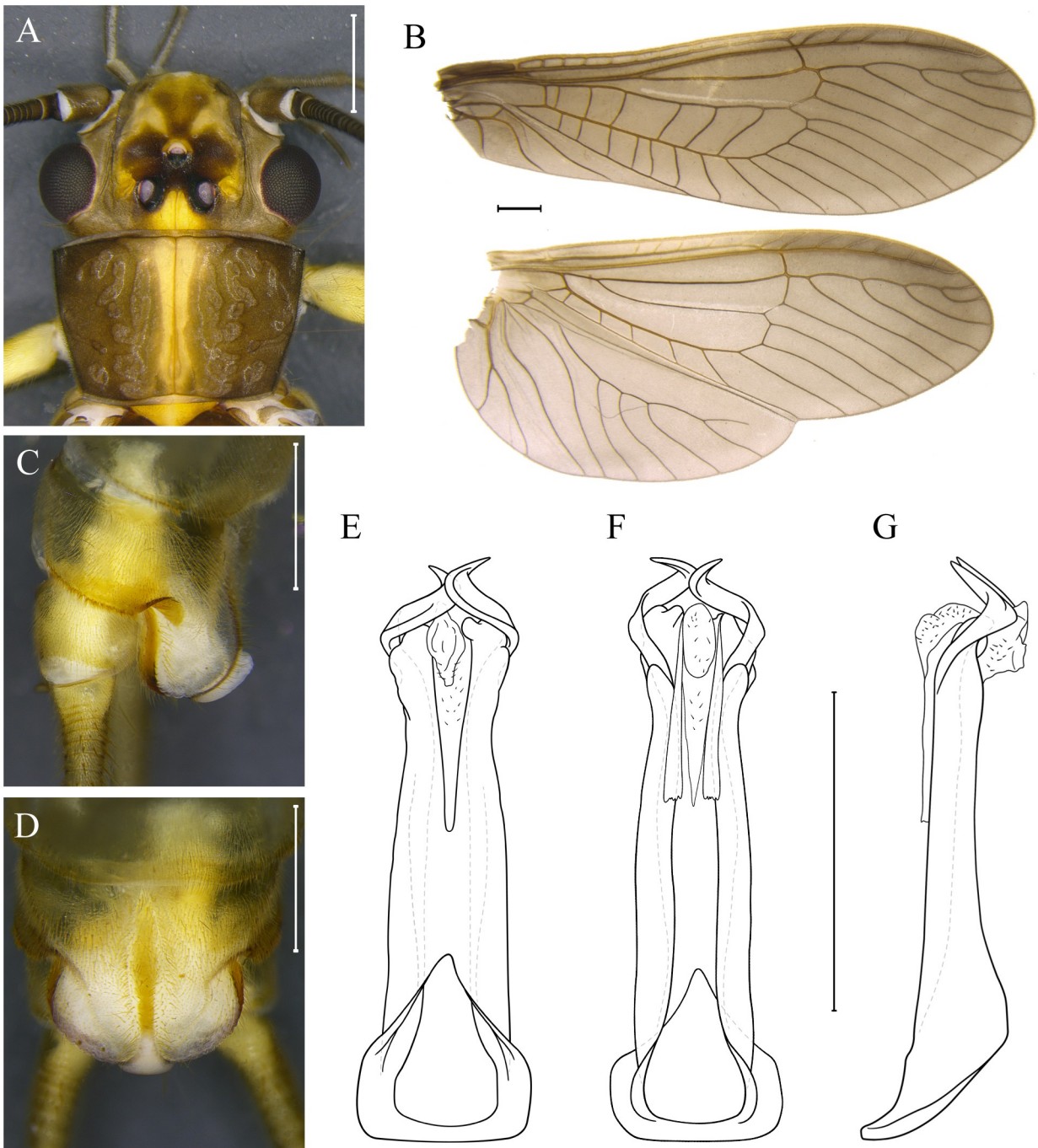

**Fig 9. *Kempnyia guarany* sp. nov.** Adult male, head and pronotum (A), wings (B) and hammer in lateral (C) and ventral views (D). Penial armature in dorsal (E), ventral (F) and lateral views (G). Scale: 1 mm.

apical sensillae with sparse minute hairs. Basal ring of armature narrow, long and flat hooks making an upward curve (Fig 9E–9G).

**Female and nymph**. Unknown.

**Remarks**. Although *K. guarany* **sp. nov**. resembles *K. neotropica* in the shape of penial armature [28, 53], the males have a larger body (16–17 mm vs. 11.5–14.2 mm) and well-

defined dark and light spots on the head and pronotum, differing from *K. neotropica* [23]. It is important to highlight that we are not talking about the color itself, but rather the pattern of spots, which have shown to be stable even among teneral and mature specimens [23]. *Kempnyia neotropica* has head and pronotum without a clearly different pattern of spots, with the exception of a longitudinal yellowish stripe on the pronotum (Fig 5). *Kempnyia guarany* **sp. nov**., on the other hand, has a clear pattern of spots on the head and pronotum, with a clearly visible M-line and light yellow spots between and outside the ocelli (Fig 9A). Additionally, in our molecular analyses we separated the two species based on the COI barcode region (Fig 1). Some species, such as *K. alterosarum* Froehlich, 1988 [21] and *K. ocellata* Froehlich, 2011 [32] resemble *K. guarany* **sp. nov**. in some aspects. However, none of them has sufficiently congruent characters with *K. guarany* **sp. nov**., which made us consider it as a new species.

**Etymology**. The name honors the Guaraní people from Brazil, who still live in indigenous areas in Santa Catarina and nearby states.

*Kempnyia tupiniquim* **sp. nov.** **Almeida, Gonçalves & Bispo** (Fig 10A–10G)
urn:lsid:zoobank.org:act:ED6D8231-8716-4CE1-A667-FD3788B1ACAE

**Material examined. BR, ES**: Santa Teresa, REBIO Augusto Ruschi, Córrego da Estrada, 27-28.xii.2017, Salles *et al*. col., 1 male (Holotype/UFVB); Córrego Bragacho, 21-22.x.2017, Salles *et al*. col., 1 male (Paratype/CLBA); 18-19.xi.2017, Salles *et al*. col., 1 male (Paratype/MZUSP).

**Measurement data**. Male (n = 3) forewing length: 10.1–11 mm (mean = 10.63 mm).

**Description**. General color whitish, with most of head and pronotum brown. Anterior ocellus absent. Frons dark brown, whitish along sides of the frons and genae; M-line light brown; lappets whitish, frontoclypeus, in front of M-line, predominantly dark brown with whitish tip; parietalia dark brown with a barely whitish longitudinal line (Fig 10A). Antennae whitish. Pronotum dark brown with rounded corners (Fig 10A). Milky wings, general color gray, whitish proximally (Fig 10B). Legs whitish, tibia darker. Cercomeres whitish.

**Male**. Tergum X pale medially and at bases of cerci, most sensilla basiconica carrot-shaped. Projection of the subgenital plate as broad as long; hammer whitish, triangle-shaped; from the base to the tip of the hammer there is a light ocher stripe; subgenital plate mostly whitish (Fig 10C and 10D). Paraprocts finger-like, with a subapical denticle; apical sensillae with sparse minute hairs. Basal ring of the penial armature broad, hooks making an almost completely round curve in the same plane (Fig 10E–10G). The membranous structures of the penial armature are large, mainly in lateral view (Fig 10G).

**Female and nymph**. Unknown.

**Remarks**. It is important to note this species was described herein based on its teneral individuals, which means that the colors shown may vary. Apparently, the general color of the mature specimen is between dark brown and black. Even though *Kempnyia tupiniquim* **sp. nov**. has a penial armature slightly similar to that of *K. vanini* [21], it is more robust, being proportionally wider. In addition, the membranous structures of the penial armature, mainly in lateral view, are larger in *K. tupiniquim* **sp. nov**. than in *K. vanini*. Despite being a membranous structure, the difference in size between them is notable. Furthermore, *Kempnyia tupiniquim* **sp. nov**. does not have the same pattern of spots observed in *K. vanini* variations, without any demarcation in the entire M-line (Fig 6). Based on the differences observed, we considered *Kempnyia tupiniquim* **sp. nov**. as a new species.

**Etymology**. The name honors the Tupiniquim people from Brazil, who still live in indigenous areas in Espírito Santo state.

*Kempnyia una* **sp. nov.** **Almeida & Bispo** (Fig 11A–11C)
urn:lsid:zoobank.org:act:3FB5D9DF-8FEB-48EC-8EC4-C2FFA198EBF8

**Material examined. BR, SP**: PEI, Rio do Carmo, bridge, 24˚18'15"S, 48˚24'31"W, 14.xii.2014, 1 female (Holotype/CLBA).

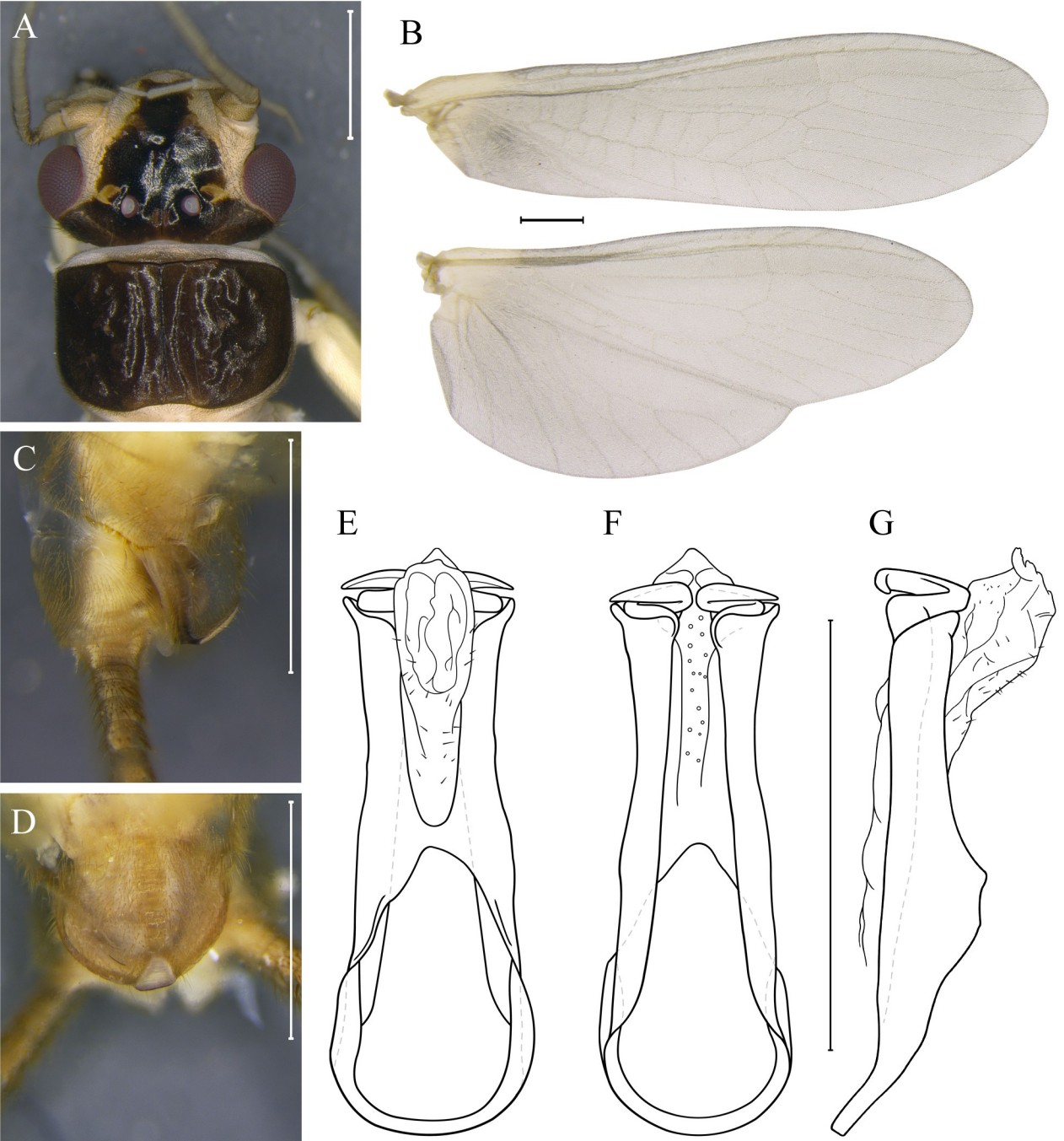

**Fig 10. *Kempnyia tupiniquim* sp. nov.** Adult male, head and pronotum (A), wings (B) and hammer in lateral (C) and ventral views (D). Penial armature in dorsal (E), ventral (F) and lateral views (G). Scale: 1 mm.

**Morphometric data**. Female (n = 1) forewing length: 9.2 mm.

**Description**. General color dark brown to yellowish. Anterior ocellus absent. Frons dark brown, light brown along sides of the frons; M-line brown and incomplete; lappets dark brown, frontoclypeus dark brown; parietalia dark brown to brown, with two transverse lighter

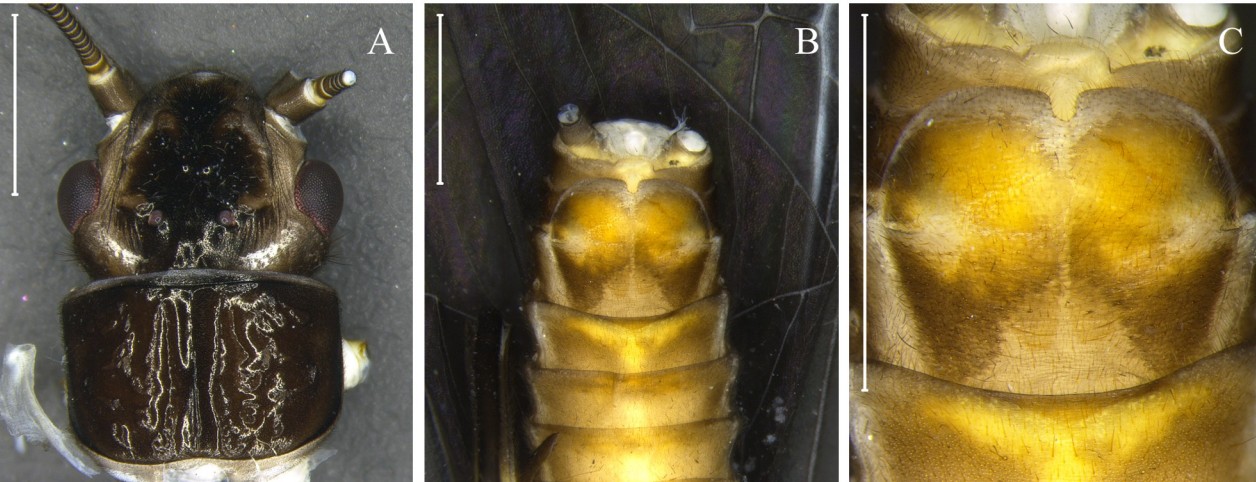

**Fig 11. *Kempnyia una* sp. nov.** Adult female, head and pronotum (A) and subgenital plate in ventral view (B–C). Scale: 1 mm.

spots that approach the compound eyes (Fig 11A). Scape of antennae brown, pedicel yellowish at the base and brown at the apex, flagellum mostly brown. Thorax dark brown, pronotum square (Fig 11A). Membrane and veins of wings dark brown, forewings with subcostal vein white in ventral view (Fig 11B). Legs dark brown to yellowish; coxae, trochanters, and base of femora and tibia dark brown; apex of femora and tibia yellowish; tarsus dark brown. Abdomen light brown. Cercomeres brown.

**Female**. Tergum X pale medially and at bases of cerci, most sensilla basiconica carrot-shaped. Subgenital plate rounded with a U-shaped chamfer in the central part of the expansion (Fig 10B and 10C).

**Male and nymph**. Unknown.

**Remarks**. Almeida & Bispo [23] originally sampled and reported the specimen studied. Because it was a female, the authors considered the specimen only as *Kempnyia* sp. After obtaining molecular data for most *Kempnyia* species, we concluded that this specimen is a new species, named *Kempnyia una* **sp. nov**. Morphologically, it has a more rounded head shape than that commonly observed among *Kempnyia* (Fig 11A). It also presents a considerable number of wrinkles near the compound eyes (Fig 11A), which is not common among specimens of the genus.

**Etymology**. Una means dark in the Tupi-Guarani language (language of Brazilian Indians).

***Kempnyia zwicki* sp. nov.** Almeida, Gonçalves & Bispo (Fig 12A–12G)
urn:lsid:zoobank.org:act:FAC77379-AAF0-4C99-9CE5-16ACAA95A533

**Material examined. BR, ES**: Santa Teresa, REBIO Augusto Ruschi, Córrego Bragacho, 18. xii.2017-17.i.2018, Salles *et al.*, 1 male (Holotype/UFVB).

**Measurement data**. Male (n = 1) forewing length: 11.5 mm.

**Description**. General color dark brown. Anterior ocellus absent. Frons dark brown, often lighter along sides of the frons and genae; M-line brown and incomplete; lappets dark brown, frontoclypeus, in front of M-line, dark brown; parietalia dark brown to brown, lighter laterally and darker in the middle near the coronal suture (Fig 12A). Scape of antennae brown, pedicel brown, flagellum mostly brown. Pronotum slightly trapezoidal and dark brown (Fig 12A). Membrane and veins of forewings brown and dark brown respectively; membrane of hindwings dark brown in the apex and lighter in anal lobe, veins dark brown (Fig 12B). Legs white to

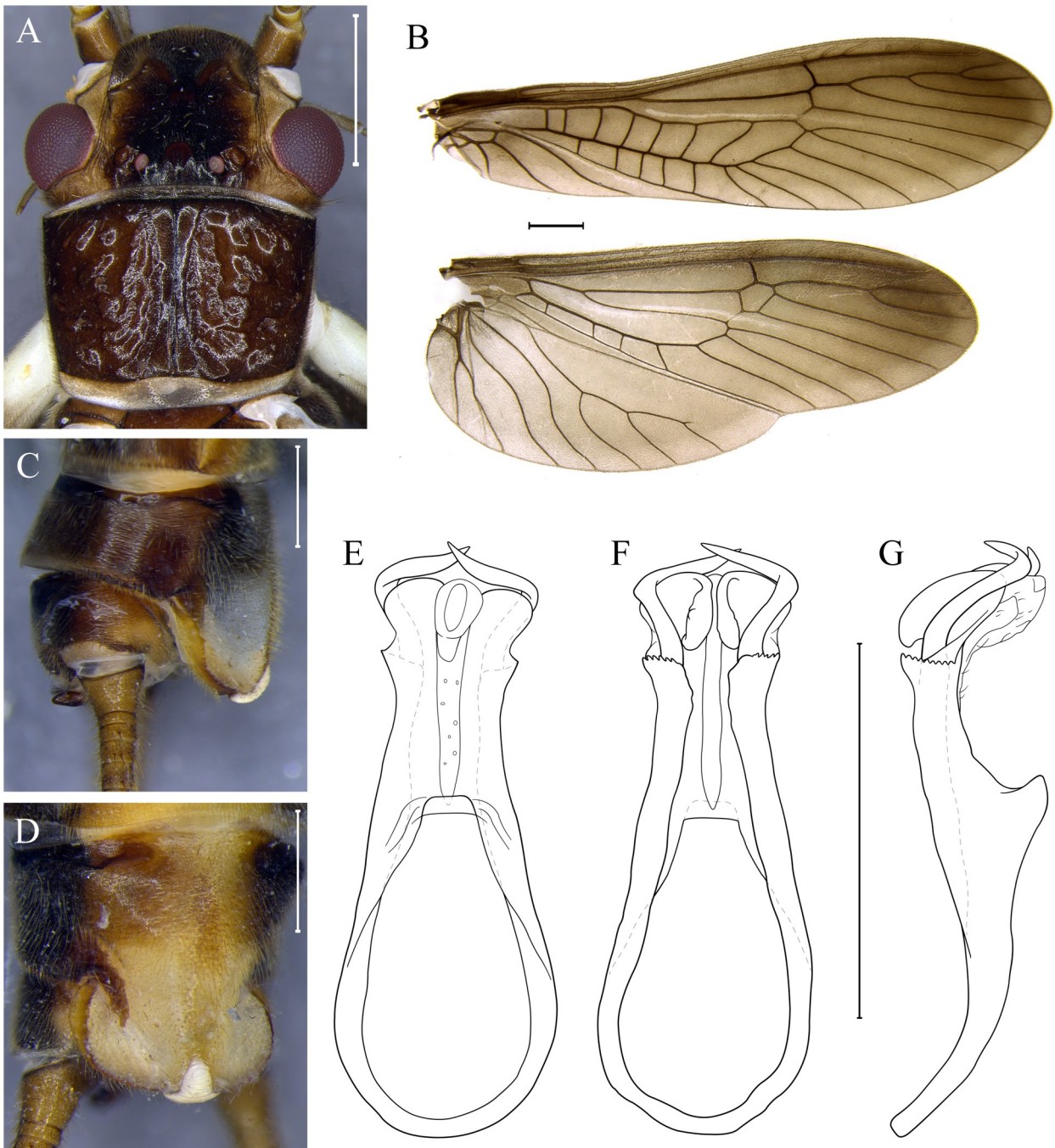

**Fig 12.** *Kempnyia zwicki* **sp. nov.** Adult male, head and pronotum (A), wings (B) and hammer in lateral (C) and ventral views (D). Penial armature in dorsal (E), ventral (F) and lateral views (G). Scale: 1 mm.

dark brown; coxae, trochanters, and base of femora and tibia white; apex of femora and tibia dark brown; tarsus dark brown. Cercomeres brown with darker apices.

**Male**. Tergum X pale medially and at bases of cerci, most sensilla basiconica carrot-shaped. Projection of the subgenital plate about twice as broad as long; T-shaped hammer white; from

the base to the tip of the hammer, there is a light ocher stripe; subgenital plate mostly whitish (Fig 12C and 12D). Paraprocts finger-like, with a subapical denticle; apical sensillae with sparse minute hairs. Basal ring of the penial armature broad, long hooks making a wide curve (Fig 12E–12G).

**Female and nymph**. Unknown.

**Remarks**. *Kempnyia zwicki* **sp. nov**. has a pattern of spots on the head and pronotum that resembles that of *K. neotropica*. Both species are almost homogeneously dark, but the shapes of penial armature are different. In lateral view, the penial armature of *K. zwicki* **sp. nov**. does not resemble that of any other species described to date.

**Etymology**. The name honors Peter Zwick, a researcher who has greatly contributed to the study of stoneflies.

## Discussion

The application of more than one source of variation to study the Perlidae family in Brazil is quite recent, with *Kempnyia* being the first genus to be investigated and the one with the largest number of DNA sequences available on GenBank. The first work using fragments of the COI mitochondrial gene to support the morphological study of species of the genus was published in 2014 [34]. In this study, the authors used COI sequences from nine species (38 specimens) of *Kempnyia* and associated adults and immatures of five species, and described *Kempnyia alterosarum* and *K. obtusa* nymphs. In fact, it is important to highlight that the molecular association of immature Plecoptera has proven to be an important tool for reducing the Hackealian shortfall, which concerns our lack of knowledge about the different life stages of a species [71]. This association is necessary since the taxonomy of the group is based entirely on adult males, and ecological and environmental monitoring studies are mainly based on immatures. Additionally, immatures can provide morphological characters that are essential to distinguish species and propose phylogenetic hypotheses.

Since the first integrative study of *Kempnyia* [34], only one work describing a new species (*K. couriae* Avelino-Capistrano, Barbosa & Takiya) and another proposing the synonymy of *K. petersorum* with *K. neotropica* have been published using DNA sequences [23, 35]. This information serves to illustrate how initial the integrative study of Plecoptera in Brazil is, with *Kempnyia* being the best studied group. In the present study, we provided 28 new COI sequences from 21 species of the genus (S1 Table), representing just over 50% of the group's diversity. Based on this sampling and supported by morphological study and automatic molecular species delimitation, we described four new species, namely, *Kempnyia guarany* **sp. nov**., *K. tupiniquim* **sp. nov**., *K. una* **sp. nov**., and *K. zwicki* **sp. nov**., increasing the diversity of the group from 36 to 40 species. Furthermore, our study brings new insights into the values of intra- and interspecific molecular divergence within the group.

The studies published until now had very high values of intraspecific molecular divergence. The minimum interspecific divergence values obtained by Avelino-Capistrano *et al.* [34], considering a sample restricted to the Macaé River Basin (Rio de Janeiro state), overlapped the maximum values of intraspecific divergences obtained in the same study. As this is a geographically restricted sampling, it was expected that the species limits would be more clearly defined, indicating some possible taxonomic noise among the identified specimens. The maximum intraspecific distance found by Avelino-Capistrano *et al.* [34] was 15.1%, a value considered high by the authors. The fact is that nowadays the understanding of the intraspecific variations of the species within the group becomes clearer due to the greater and more diverse existing number of COI sequences. In this context, our study found a maximum intraspecific variation of 1.4% between *K. gracilenta* (DP82) and (DP83), and a minimum interspecific variation of

8.1% between *Kempnyia mirim* (DP269) and *K. tamoya* (DP314)–values very different from those reported previously. Therefore, extremely high intraspecific divergences, such as the 14% found for *Kempnyia colossica* by Almeida and Bispo [23], suggest a species complex. Gradually, with this and other studies [23, 34, 35], the taxonomic problems and possible paths to their solutions become clearer.

Intra- and interspecific variation values alone do not define rules, as they may vary from group to group and also suffer from some bias in terms of the sampling effort. Therefore, the ideal is that the automatic molecular delimitation of species be carried out by a set of automated methods and aiming at a consensus between them, as we did in the present study. In addition to supporting the description of new species, our study demonstrated that the species delimitation methods considered simple were sufficient and efficient for the delimitation of *Kempnyia*. This is probably because these insects have a very restricted distribution, occurring mainly in mountain streams in preserved areas. Despite this, it is expected that the molecular intraspecific divergence values observed in the present study will increase as new sequences are added and sampling is expanded geographically. This would imply a narrower interval between intra- and interspecific divergences.

Finally, the present study opens a new chapter in the history of studies on *Kempnyia*, expanding the number of sequences on GenBank and reinforcing the validity of using DNA sequences as a source of additional information for morphological studies. We hope that this paper will stimulate new studies on *Kempnyia*, advancing the understanding of the molecular and morphological limits among the species of the genus. This will certainly enable discussions that are even more complex, including the search for conservation strategies for these insects, which are highly vulnerable to environmental modifications caused by human activities.

## Supporting information

**S1 Table. Voucher codes.** Specimen vouchers with respective identification, collecting local and GenBank accession codes of COI sequences.
(DOCX)

**S2 Table. Kimura-2-parameter (K2P) divergences of COI sequences.**
(XLS)

## Acknowledgments

The authors thank to all collectors and their team for sampling the material.

## Author Contributions

**Conceptualization:** Lucas Henrique de Almeida, Pitágoras da Conceição Bispo.

**Data curation:** Lucas Henrique de Almeida, Maísa de Carvalho Gonçalves, Pitágoras da Conceição Bispo.

**Formal analysis:** Lucas Henrique de Almeida, Maísa de Carvalho Gonçalves.

**Funding acquisition:** Pitágoras da Conceição Bispo.

**Investigation:** Lucas Henrique de Almeida, Maísa de Carvalho Gonçalves, Pitágoras da Conceição Bispo.

**Methodology:** Lucas Henrique de Almeida, Maísa de Carvalho Gonçalves.

**Project administration:** Lucas Henrique de Almeida, Pitágoras da Conceição Bispo.

**Resources:** Lucas Henrique de Almeida, Pitágoras da Conceição Bispo.

**Software:** Lucas Henrique de Almeida.

**Supervision:** Lucas Henrique de Almeida, Pitágoras da Conceição Bispo.

**Validation:** Lucas Henrique de Almeida, Pitágoras da Conceição Bispo.

**Visualization:** Lucas Henrique de Almeida, Pitágoras da Conceição Bispo.

**Writing – original draft:** Lucas Henrique de Almeida, Maísa de Carvalho Gonçalves.

**Writing – review & editing:** Lucas Henrique de Almeida, Maísa de Carvalho Gonçalves, Pitágoras da Conceição Bispo.

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
