## [Decision Letter · Decision Letter 0]

11 Apr 2024

PONE-D-24-09214An integrative approach to the study of Kempnyia Klapálek, 1914 (Plecoptera: Perlidae) from Brazil: Support for the description of four new species and a basis for future studies.PLOS ONE

Dear Dr Almeida,

Thank you for submitting your manuscript to PLOS ONE. After careful consideration, we feel that it has merit but does not fully meet PLOS ONE’s publication criteria as it currently stands. Therefore, we invite you to submit a revised version of the manuscript that addresses the points raised during the review process.

We look forward to receiving your revised manuscript.

Kind regards,

James Lee Crainey, Ph.D.

Academic Editor

PLOS ONE

Journal Requirements:

“The authors thank to all collectors and their team for sampling the material and to FAPESP (Research Foundation of São Paulo, grants 2019/22833-0 and BIOTA 2021/05986-8) and CNPq (National Council for Scientific and Technological Development, grant PROTAX 441119/2020-4) for the financial support. LHA and MCG thank to CAPES (Coordination for the Improvement of Higher Education Personnel) for the scholarship. LHA thanks to UNESP (São Paulo State University) and PROTAX-CNPq (grant 150032/2024-2) for the postdoctoral fellowship. PCB thanks to CNPq (grant 306400/2022-7) for the Research Productivity Fellowship. “

“All authors: Research Foundation of São Paulo (FAPESP) (grants 2019/22833-0 and BIOTA 2021/05986-8) and National Council for Scientific and Technological Development (CNPq) (grant PROTAX 441119/2020-4).

LHA and MCG: Coordination for the Improvement of Higher Education Personnel (CAPES).

LHA: São Paulo State University (UNESP) and PROTAX-CNPq (grant 150032/2024-2).

PCB: CNPq (grant 306400/2022-7).”

4. We note that Figures 2-8 in your submission contain [map/satellite] images which may be copyrighted. All PLOS content is published under the Creative Commons Attribution License (CC BY 4.0), which means that the manuscript, images, and Supporting Information files will be freely available online, and any third party is permitted to access, download, copy, distribute, and use these materials in any way, even commercially, with proper attribution. For these reasons, we cannot publish previously copyrighted maps or satellite images created using proprietary data, such as Google software (Google Maps, Street View, and Earth). For more information, see our copyright guidelines: http://journals.plos.org/plosone/s/licenses-and-copyright.

 a. You may seek permission from the original copyright holder of Figures 2-8 to publish the content specifically under the CC BY 4.0 license. 

Natural Earth (public domain): http://www.naturalearthdata.com/.

5. Please take this opportunity to be sure you have met all of our guidelines for new species. For proper registration of a new zoological taxon, we require two specific statements to be included in your manuscript.

a.        In the Results section, the globally unique identifier (GUID), currently in the form of a Life Science Identifier (LSID), should be listed under the new species name, for example:

Anochetus boltoni Fisher sp. nov. urn:lsid:zoobank.org:act:B6C072CF-1CA6-40C7-8396-534E91EF7FBB

Another LSID for the manuscript itself should also appear within the Nomenclature statement. You will need to contact Zoobank (zoobank.org/About) to obtain a GUID (LSID). You should receive one LSID for your manuscript and a separate, unique LSID for the new species.

b.        Please also insert the following text into the Methods section, in a sub-section to be called "Nomenclatural Acts":

The electronic edition of this article conforms to the requirements of the amended International Code of Zoological Nomenclature, and hence the new names contained herein are available under that Code from the electronic edition of this article. This published work and the nomenclatural acts it contains have been registered in ZooBank, the online registration system for the ICZN. The ZooBank LSIDs (Life Science Identifiers) can be resolved and the associated information viewed through any standard web browser by appending the LSID to the prefix "" ext-link-type="uri" xlink:type="simple">http://zoobank.org/". The LSID for this publication is: urn:lsid:zoobank.org:pub: XXXXXXX. The electronic edition of this work was published in a journal with an ISSN, and has been archived and is available from the following digital repositories: PubMed Central, LOCKSS [author to insert any additional repositories].

All PLOS ONE articles are deposited in PubMed Central and LOCKSS. If your institute, or those of your co-authors, has its own repository, we recommend that you also deposit the published online article there and include the name in your article.

Following a recent ruling by the International Commission on Zoological Nomenclature, electronic journals are now a valid format for publication of new zoological taxa. In order to ensure the valid publication of your new species, please be sure to include the updated version of Nomenclatural Acts (above). A complete explanation of our guidelines for publishing new species can be found on our website: http://www.plosone.org/static/guidelines#zoological.

Reviewers' comments:

Reviewer's Responses to Questions

**Comments to the Author**

1. Is the manuscript technically sound, and do the data support the conclusions?

Reviewer #1: Yes

Reviewer #2: Yes

2. Has the statistical analysis been performed appropriately and rigorously? 

Reviewer #1: Yes

Reviewer #2: I Don't Know

3. Have the authors made all data underlying the findings in their manuscript fully available?

Reviewer #1: No

Reviewer #2: Yes

4. Is the manuscript presented in an intelligible fashion and written in standard English?

Reviewer #1: Yes

Reviewer #2: Yes

5. Review Comments to the Author

Reviewer #1: I finalized my evaluation on the manuscript "An integrative approach to the study of Kempnyia Klapálek, 1914 (Plecoptera: Perlidae) from Brazil: Support for the description of four new species and a basis for future studies."

I would like to praise the authors for the consistency presented in all topics covered in the Introduction, Material and Methods, Results and Discussion, demonstrating the importance of the study.

Some suggestions

In the methodology, authors are strongly requested to deposit their sequences in a database, Genbank or BOLD and include public access in the final version of the manuscript.

In ABGD, What prior was considered for intraspecific divergences for the study?

In results, the description of the species is well defined and discussed, presenting the consistency of the information.

I suggest that all requested information be included in the final version before publication.

Reviewer #2: Thankyou for providing such a detailed and well-written method. My comments concern the results and discussion:

184: Is there a reference for the previous morphological identification? How did you decide where to place the new species on the nominal tree in Fig 1? Could provide more detail for the morphology-based taxonomy.

755: "The studies published until THEN". Until when? Do you mean NOW? This section isn't completely clear.

764-771: It is unclear how the values you highlight, shed light on the problem with Kempnyia colossica. Also restate that that the interspecific variation of 8.1% is the minimum interspecific variation. This section could be clearer (without the reader having to go back though the results).

6. PLOS authors have the option to publish the peer review history of their article (what does this mean?). If published, this will include your full peer review and any attached files.

Reviewer #1: No

Reviewer #2: No

---

## [Author Response · Author response to Decision Letter 0]

3 May 2024

Rebuttal Letter

Dear Editor

We would like to submit the revised manuscript “An integrative approach to the study of Kempnyia Klapálek, 1914 (Plecoptera: Perlidae) from Brazil: Support for the description of four new species and a basis for future studies” to continue the publication process in “Plos One”.

We would like to thank the Editor and the two reviewers for their suggestions. We considered all the suggestions given by the reviewers and all the requirements questioned by the Editor. We would just like to highlight some important points:

1. About the presence of financing in the acknowledgments: This has been corrected and the information on financing obtained is correctly entered into the system.

2. 2. About the maps: The shapefile used was obtained from SimpleMappr, an online and free-to-use platform. In any case, we added SimpleMappr to the Materials and Methods and added to this Rebuttal Letter the excerpt that qualifies SimpleMappr as a free platform, which is present on the platform creator's website (https://www.simplemappr.net/#tabs=6).

“All versions of SimpleMappr map data found on this website are in the Public Domain. You may use the maps in any manner, including modifying the content and design, electronic dissemination, and offset printing. The primary author, David P. Shorthouse has waived all copyright, related or neighboring rights, and financial claim to the maps and invites you to use them for personal, educational, and commercial purposes. No permission is needed to use SimpleMappr.” 

 Best regards

Authors

---

## [Decision Letter · Decision Letter 1]

21 May 2024

PONE-D-24-09214R1An integrative approach to the study of Kempnyia Klapálek, 1914 (Plecoptera: Perlidae) from Brazil: Support for the description of four new species and a basis for future studies.PLOS ONE

Dear Dr. de Almeida,

Thank you for submitting your manuscript to PLOS ONE. After careful consideration, we feel that it has merit but does not fully meet PLOS ONE’s publication criteria as it currently stands. Therefore, we invite you to submit a revised version of the manuscript that addresses the points raised during the review process.

Congratulations on your fine manuscript! The reviewers are both satisfied that your manuscript is ready for publication and my concerns are limited to the written English and are very minor indeed. Please take a look at the very light copy-editing I have performed on your manuscript and make sure that it has not changed the meaning of anything. There is no need for you to accept all the changes I am suggesting, but please re-word the sentences I have changed I have changed if you don´t like what I have written. I am giving you a month to make these changes, but I honestly don´t think you´ll need more than an afternoon.

We look forward to receiving your revised manuscript.

Kind regards,

James Lee Crainey, Ph.D.

Academic Editor

PLOS ONE

Journal Requirements:

Additional Editor Comments:

Congratulations on your fine manuscript! The reviewers are both satisfied that your manuscript is ready for publication and my concerns are limited to the written English and are very minor indeed. Please take a look at the very light copy-editing I have performed on your manuscript and make sure that it has not changed the meaning of anything. There is no need for you to accept all the changes I am suggesting, but please re-word the sentences I have changed I have changed if you don´t like what I have written. I am giving you a month to make these changes, but I honestly don´t think you´ll need more than an afternoon.

Reviewers' comments:

Reviewer's Responses to Questions

**Comments to the Author**

1. If the authors have adequately addressed your comments raised in a previous round of review and you feel that this manuscript is now acceptable for publication, you may indicate that here to bypass the “Comments to the Author” section, enter your conflict of interest statement in the “Confidential to Editor” section, and submit your "Accept" recommendation.

Reviewer #1: All comments have been addressed

2. Is the manuscript technically sound, and do the data support the conclusions?

Reviewer #1: Yes

3. Has the statistical analysis been performed appropriately and rigorously? 

Reviewer #1: Yes

4. Have the authors made all data underlying the findings in their manuscript fully available?

Reviewer #1: Yes

5. Is the manuscript presented in an intelligible fashion and written in standard English?

Reviewer #1: Yes

6. Review Comments to the Author

Reviewer #1: Dear authors

After reading the new version of the manuscript, I have no further comments to add. My decision is to consider the study for publication.

7. PLOS authors have the option to publish the peer review history of their article (what does this mean?). If published, this will include your full peer review and any attached files.

Reviewer #1: No

---

## [Author Response · Author response to Decision Letter 1]

24 May 2024

Rebuttal Letter

Dear Editor

We would like to submit the revised manuscript “An integrative approach to the study of Kempnyia Klapálek, 1914 (Plecoptera: Perlidae) from Brazil: Support for the description of four new species and a basis for future studies” to continue the publication process in “Plos One”.

We would like to thank the Editor and the two reviewers for their suggestions. We considered all the suggestions given by the Reviewers and Editor. 

 Best regards

Authors

---

## [Editor Report · Decision Letter 2]

5 Jun 2024

An integrative approach to the study of Kempnyia Klapálek, 1914 (Plecoptera: Perlidae) from Brazil: Support for the description of four new species and a basis for future studies.

PONE-D-24-09214R2

Dear Dr. Almeida,

We’re pleased to inform you that your manuscript has been judged scientifically suitable for publication and will be formally accepted for publication once it meets all outstanding technical requirements.

Kind regards,

James Lee Crainey, Ph.D.

Academic Editor

PLOS ONE
---

## [Editor Report · Acceptance letter]

21 Jun 2024

PONE-D-24-09214R2 

PLOS ONE

Dear Dr. de Almeida, 

I'm pleased to inform you that your manuscript has been deemed suitable for publication in PLOS ONE. Congratulations! Your manuscript is now being handed over to our production team.

Kind regards, 

on behalf of

Dr. James Lee Crainey 

Academic Editor

PLOS ONE